# Analysis of the lightning production of convective cells

Jordi Figueras i Ventura[1], Nicolau Pineda[2,3], Nikola Besic[1], Jacopo Grazioli[1], Alessandro Hering[1], Oscar A. van der Velde[3], David Romero[3], Antonio Sunjerga[4], Amirhossein Mostajabi[4], Mohammad Azadifar[5], Marcos Rubinstein[5], Joan Montanyà[3], Urs Germann[1], and Farhad Rachidi[4]

[1]MeteoSwiss, Locarno, Switzerland
[2]Meteorological Service of Catalonia, Barcelona, Spain
[3]Universitat Politècnica de Catalunya UPC, Terrassa, Spain
[4]École Polytechnique Fédérale de Lausanne EPFL, Lausanne, Switzerland
[5]Haute Ecole Spécialisée de Suisse occidentale HES-SO, Yverdon-les-bains, Switzerland

**Correspondence:** Jordi Figueras i Ventura (jordi.figuerasiventura@meteoswiss.ch)

**Abstract.** This paper presents an analysis of the lightning production of convective cells. The cells were detected by the MeteoSwiss Thunderstorms Radar Tracking (TRT) algorithm in the course of a lightning measurement campaign that took place in the summer of 2017 in the area surrounding the Säntis mountain, in the northeastern part of Switzerland. For this campaign, and for the first time in the Alps, a Lightning Mapping Array (LMA) was deployed. In the first part of the paper, we examine the relationship between the intra-cloud (IC) and cloud-to-ground (CG) activity and the cell severity, as derived by the TRT algorithm, of a large dataset of cells gathered during the campaign. We also propose and analyze the performance of a new metric to quantify lightning intensity, the rimed particles column (RPC) height and base altitude. In the second part, we focus on two of the most severe cells detected during the campaign that produced significantly different outcomes in terms of lightning activity. The paper shows that the newly proposed metric (RPC) seems to be a very promising predictor of lightning activity, particularly for IC flashes. Future lightning nowcasting algorithms should be probabilistic in nature and incorporate the polarimetric properties of the convective cells as well as the lightning climatology.

*Copyright statement.* TEXT

## 1 Introduction

Current meteorological warnings are strictly based on meteorological features. However there is a societal need for more impact-based meteorological warnings (WMO Public Weather Services Programme, 2015). Impact-based warnings would allow more effective preventive actions and an optimization of the deployment of the emergency services in the most critical areas. In order to issue impact-based warnings though, the type of natural hazard has to be identified and forecasted with precision since the required preventive and response actions to be taken for each meteorological phenomenon may be different. Lightning activity nowcasting systems are still lacking in precision and reliability. Warnings on lightning activity would be very valuable in many areas such as the organization of outdoors events, for the safety in areas of high lightning risk (e.g.

warnings for maintenance personnel in wind farms or tall structures), airport warning systems, etc. Moreover, lightning jumps (Schultz et al., 2009) have been associated to severe weather intensification. If lightning activity can be reliably nowcasted, it can therefore have a positive impact in the forecasting of other phenomena.

Since 2004, the Thunderstorms Radar Tracking (TRT) algorithm is operational at MeteoSwiss (Hering et al., 2004). The algorithm identifies, tracks and characterizes convective cells in real-time with 5-min resolution using data from the operational C-band polarimetric weather radar network. The TRT algorithm ranks the identified cells in 5 categories according to their severity. The position of each cell is extrapolated 60 minutes into the future according to its moving direction and speed. This algorithm is part of the semi-automatized short-term severe weather warning system operated by MeteoSwiss (Hering et al., 2015). The warning level is based on the cell rank category: weak (for which no warning is issued), developing, moderate, severe and very severe. Implicit in such categories there is information of expected precipitation intensity, wind gusts and hail. Explicit lightning activity warnings are generated by a separate system. These warnings are primarily based on the observation of lightning activity in the area surrounding the airports and issued for airports only. An explicit forecast of the lightning activity within the context of convective cells would thus be an appreciated enhancement. It should be mentioned that there have already been some attempts at lightning nowcasting based on single-polarization radar products (e.g., Mosier et al., 2011; Seroka et al., 2012).

Lightning activity can be roughly divided into intra- or inter-cloud (IC) and cloud-to-ground (CG) categories. CG flashes can be reliably detected and located using continental-scale networks of low frequency sensors such as the EUCLID network in Europe. Such networks, though, are not as efficient at detecting IC flashes (Poelman et al., 2016). Lightning mapping arrays (LMAs), on the other hand, provide 3D information of the discharge path with a much higher temporal and spatial resolution. An LMA is a network of VHF sensors that measure the arrival time of VHF radiation and uses the information to estimate the location of the intracloud channel sources (Proctor, 1971). The return stroke in CG strokes does not produce such intense VHF emissions as lightning leaders and therefore it is not evident to classify flashes as CG. Hence, both types of networks are complementary.

Polarimetric weather radar networks provide extremely useful information about the state of the atmosphere in precipitating systems, particularly in convective ones. They offer a large coverage of the 3D cube of the atmosphere (from altitudes close to ground to about $20000\,\mathrm{m\,MSL}$), with high spatial resolution (on the order of hundreds of m) and relatively high temporal resolution, typically $5\,\mathrm{min}$ for operational systems. Numerous studies have been conducted combining information from both lightning detectors and polarimetric radars. Furthermore, various direct and indirect links between polarimetric signatures and lightning activity have been reported in the past (see the introduction of Figueras i Ventura et al. (2019) for an overview).

From early on, the scientific community has realized that there is a strong link between the distribution of different hydrometeors, particularly their vertical distribution, and the lightning activity of the system. Consequently, significant effort has been placed in characterizing the vertical structure of precipitation, either from ground-based or from satellite measurements. For example, Marra et al. (2017) reported on a violent hailstorm in the Gulf of Naples that produced over 37000 strokes in $5\,\mathrm{h}$ and hailstones with a maximum size in the order of 7-10 $\mathrm{cm}$. The storm happened to be observed by an overpass of the Global Precipitation Measurement Mission Core Observatory (GPM-CO). The on-board Dual-frequency Precipitation Radar

(DPR) observed 40 dBZ echo top heights of up to 14 km, indicative of strong updraft, and observations from both DPR and ground-based radar indicated the presence of large graupel/hail particles at the time where there was an increase in the intra-cloud positive stroke fraction as observed by the lightning network LINET. Buiat et al. (2017) analyzed a total of 12 convective events over Italy that were observed by the Cloud Profiling Radar (CPR) on board of the CloudSat satellite. A high correlation was found between the number of strokes, detected by LINET, and the vertical distribution of ice particles, as estimated from the CPR products. A high ice water content and large effective radius of the ice particles favoured and increase in CG stroke occurrence.

Phase-array radar technology may have the potential to greatly improve the characterization of the precipitating systems where lightning occurs thanks to the fact that their fast scanning may better capture the evolution of the rapidly evolving convective cells. In this context a study by Emersic et al. (2011) is noteworthy. The authors examined data of a hail-producing storm on 15 August 2006 in Oklahoma obtained by the National Weather Radar Testbed Phased-Array Radar. Interestingly, it was found that, while there was a first surge in lightning when the storm's updraft first intensified, a second updraft later on actually coincided with a decrease in total flash rate. The authors attribute this behaviour to the formation of wet hail preventing hydrometeor charging.

Another study, by Carey and Rutledge (2000), reported on a tropical convective complex over the Tiwi Islands on 28 November 1995, observed by a C-band polarimetric radar. It is noteworthy that in its initial phase the storm was dominated by warm rain processes. During this phase, no-significant lightning activity was detected despite of the considerable rainfall rate. At a later stage, a gust front contributed to the formation of more intense convective complexes that were dominated by mixed-phase precipitation processes. It was during this phase that all the lightning occurred. The presence of rimed particles was thus, again, decisive in the electrification of the storm. Finally, noteworthy for its extent, Wapler (2017) analyzed 600 hailstorms that occurred over a period of 8 years in various parts of Germany. Among other findings it is shown that hail bearing storms are far more likely to have a high stroke rate than regular storms. The stroke rate tends to increase minutes before hail reaches the ground and decreases after that.

In this paper, we analyze a large number of convective cells detected during a lightning measurement campaign that took place in the summer of 2017 in the area surrounding the Säntis mountain, northeast of Switzerland. For this campaign, and for the first time in the Alps, an LMA was deployed. The main goals of this study are:

- To explore the relationship between the TRT cell rank and the lightning activity (both intra-cloud IC and cloud-to-ground CG).

- To present and evaluate the performance of a new indicator of possible lightning activity based on polarimetric radar data, the rimed particles column (RPC) base altitude and height.

- To examine in detail and compare the characteristics of two convective cells that reached similar levels of severity but produced very different outputs in terms of lightning activity.

The paper is organized as follows: Section 2 provides a brief overview of the Säntis measurement campaign and the instrumentation and methods used for this study. Section 3 contains a statistical analysis of the entire dataset. Section 4 presents a detailed analysis of two severe convective cells. General conclusions and recommendations are given in section 5.

## 2 Instrumentation and methods

The Säntis measurement campaign was a joint venture between the Electromagnetic Compatibility Laboratory (EMC LAB) of the Swiss Federal Institute of Technology in Lausanne (EPFL), the Institute for Information and Communication Technologies of the University of Applied Sicences of Western Switzerland (HES-SO), the Lightning Research Group (LRG) of the Technical University of Catalonia, the Meteorological Service of Catalonia (meteo.cat) and the Radar Satellite and Nowcasting Division of the Federal Office of Meteorology and Climatology MeteoSwiss. The campaign took place in the summer of 2017. The main objective of the campaign was to study the atmospheric conditions leading to lightning production in the vicinity of the Säntis telecommunications tower, with particular focus on the upward lightning discharges initiated by the tower itself. In this study, though, we focus on the lightning production of convective cells regardless of their origin. The 124 m tall telecommunications tower is situated on top of the Säntis mountain (47.2429°N, 9.3393°E, 2502 m MSL), in the Sankt Gallen Canton, in the north-eastern part of Switzerland. The main instruments of the campaign were in-situ measurements on the tower, including lightning current and static electric field measurements, a Lightning Mapping Array (LMA) network and a polarimetric Doppler weather radar network. The area covered by the campaign and the location of the instrumentation can be seen in Fig. 1. In the following, a brief description of the instrumentation used during the campaign is provided.

### 2.1 Radar data

#### 2.1.1 The operational MeteoSwiss Doppler polarimetric weather radar network

MeteoSwiss owns and operates a network of 5 C-band, Doppler polarimetric weather radars. The network was recently renewed within the project Rad4Alp, which was concluded in 2016 (Germann et al., 2015). The 5 systems have identical specifications and modes of operation. The scanning strategy consists of 20 horizontal scans with elevations ranging from -0.2° to 40°, repeated every 5 min. The elevations are inter-leaved: every 2.5 min a half-volume of 10 elevations from top to bottom is concluded. A very short pulse of 0.5 μs is used to obtain data with a range resolution of 83.3 m with angular resolution of 1°. IQ data are processed on-site using standard techniques (e.g., Doviak and Zrnic, 2006) to obtain the basic polarimetric moments, i.e., reflectivity (horizontal $Z_h$ and vertical $Z_v$), differential reflectivity ($Z_{dr}$), co-polar correlation coefficient ($\rho_{hv}$) and raw co-polar differential phase ($\psi_{dp}$) as well as Doppler moments. These basic moments are transmitted to a central server. The operational data processing involves a clutter detection using a sophisticated decision tree filter (DT-filter) and a reduction of the resolution to 500 m by averaging 6 consecutive gates (only clutter-free ones). From the low resolution polarimetric moments all subsequent products are generated. For the measurement campaign, data from the Albis radar (47.2843°N 8.5120°E, 938 m MSL), situated 63 km east of the Säntis tower, were used (see Fig. 1).

#### 2.1.2 The Thunderstorms Radar Tracking (TRT) algorithm

Since 2004, MeteoSwiss operates the automatic Thunderstorms Radar Tracking algorithm TRT (Hering et al., 2004). The algorithm identifies, tracks and characterizes convective cells in real-time with 5 minutes resolution using as input the reflectivity

data from the 3D radar composite (1 km horizontal resolution and 200 m vertical resolution up to 18 km MSL) and the iso-0°C from NWP models and lightning data from the EUCLID network as auxiliary parameters. The detection of the cells is based on a dynamic thresholding scheme applied on the reflectivity data of multiple-radar composites. For each radar pixel, the columnar maximum reflectivity is defined. A cell is defined as a connected area of radar pixels larger than a given area threshold and whose reflectivity exceeds an adaptive detection threshold. The detection threshold is chosen so that 1) it is above a minimum, 2) the difference between the maximum value of the cell and a member pixel is above a certain threshold, 3) the area covered by the cell is smaller than a certain threshold. The cells are currently classified in five categories (weak [0-1.2[, developing [1.2-1.5[, moderate [1.5-2.5[, severe[2.5-3.5[ and very severe [3.5-4.0[) according to their severity ranking (Hering et al., 2008). The categorization is performed by examining the values of vertical integrated liquid content (VIL [$\mathrm{kg \cdot m^{-2}}$]), median cell echo top 45 dBZ ($ET45m$ [km]), maximum cell reflectivity ($dBZmax$ [dBZ]) and the extension of the area of the cell having a reflectivity above 55 dBZ ($area55dBZ$ [$\mathrm{km^2}$]):

$$RANK = (2 \cdot VIL + 2 \cdot ET45m + dBZmax + 2 \cdot area55dBZ)/7.0 \tag{1}$$

The tracking of the cell is performed by searching areas of overlap between cells of two consecutive images (the current time *t* and the previous one *t-t₀*). The cells at *t-t₀* are advected using the estimated cell velocity and the overlapping area between the advected cell and each of the cells detected at time *t* is computed. Cells with the maximum overlapping area, provided that it is above a minimum threshold, are considered to be the same and given the same unique ID. The velocity of each cell is computed by examining the displacement of the cell center between two consecutive images and taking the weighted average of all previous velocities, calculated recursively with a decreasing weight, or by a cross-correlation technique if no displacement of the cell centres can be found. Once the cell has been properly identified, various parameters are computed to better characterize the cell. The parameters computed are summarized in Table 1. Among them, the number of lightning strokes is computed by counting all the strokes detected by the EUCLID network within the area covered by the cell and within the time resolution of the TRT cell (i.e. within the 5 min prior to the current time stamp). All these parameters are stored in a file (one file per radar image every 5 minutes). This information, accessible also in real-time, is very useful to study the evolution of the convective cells.

### 2.1.3 Additional radar data processing and analysis tools

A specific non-operational processing was performed on radar data obtained in real time during the campaign. The processing was performed using the Python-based open source software Pyrad/Py-ART (Figueras i Ventura et al., 2017). Detailed information on the processing is provided in Figueras i Ventura et al. (2019). It is sufficient to mention here that at the end of the processing, high resolution (83.3 m) clutter-free volumes of attenuation-corrected horizontal reflectivity $Z_h$ and differential refletivity $Z_{dr}$, co-polar correlation coefficient $\rho_{hv}$, specific differential phase $K_{dp}$, air temperature from the Numerical Weather Prediction (NWP) model COSMO-1 (see http://www.cosmo-model.org/) re-sampled at the radar resolution and the dominant hydrometeor type at each range gate were obtained. The hydrometeor classification is described in Besic et al. (2016) and it provides the following hydrometeor classes: aggregates (AG), ice crystals (CR), light rain (LR), rimed particles (RP),

rain (RN), vertically-oriented ice crystals (VI), wet snow (WS), melting hail (MH), ice hail/high density graupel (IH) and no classification (No valid radar data) NC. These data were used in the subsequent analysis.

Within the radar data processing tool Pyrad, a TRT trajectory function has been implemented. This function uses the cell footprints defined by the TRT algorithm to extract all the (3D) radar volume data contained within its boundaries. The 3D volume corresponds to the vertical extrapolation of the 2D cell footprint of the TRT, i.e. its section is invariant with height. Out of this dataset, several products can be generated. For example, one product computes histograms over the entire vertical data column, another computes various quantiles, a third product, the so-called vertical profile, obtains the vertical profile of user-defined statistics (mean, median, mode, etc.) at prescribed height levels, etc. More specifically, the vertical profile is constructed by taking statistics of all the valid data at a particular height interval. In the cases shown in this study, the height resolution was set to 250 m, e.g. all data at altitudes ranging from 0 to 250 m were used to compute statistics valid for that height interval. Notice that depending on the size of the cell and its position with respect to the radar, there may be height intervals where no data is present simply because no radar beam covers the sampled volume.

A similar rationale is used to extract data obtained by the LMA within the TRT cell footprint. Any LMA-detected lightning is assigned to the cell if it has sources located within the TRT cell area (regardless of its origin) within the time resolution of the TRT algorithm (i.e. in the last 5 min from the current TRT time stamp). Out of the resultant data, products such as the total number of flashes, total number of sources and vertical profile of the number of flashes and sources can be computed.

### 2.1.4 Rimed particles column computation

Out of the hydrometeor-classification cell profile, constructed by taking the mode at each height level of 250 m resolution, we compute the rimed particles column. We consider the base of the column as the height of the bottom of the lowest height level where rimed particles or hail are predominant and the top of the column as the height of the top of the highest height level where those species are predominant. The RPC height is therefore the difference between those altitudes. The possibility that height levels within this column have other predominant hydrometeors is neglected since we assume that isolated rimed particle areas cannot exist so in any case a significant if not dominant proportion of hydrometeors would be rimed particles.

The RPC has several sources of uncertainty. In the first place, the nominal resolution of the column is equal to the resolution of the cell profile. Hence, in our particular case, columns less than 250 m high will not be detected. The effective resolution though depends on the length of the volume scanned by the radar, which is determined by the radar beamwidth. With a beamwidth of $1°$, the effective resolution exceeds 250 m at approximately 14 km range. Secondly, it is dependent on the precision of the hydrometeor classification. Generally speaking, in areas with good visibility, rimed particle columns have a clear signature that allows a good differentiation between them and other solid species such as ice crystals and aggregates, provided that precipitation-induced attenuation has been sufficiently accounted for and the radar is reasonably well calibrated. Classification may be more prone to errors in transitioning areas, particularly close to the melting layer, where no hydrometeor can be considered dominant. Consequently, the uncertainty is higher in determining the base of the RPC. A third source of uncertainty is related to the time resolution of the radar scan. In our case, it takes 5 min to sample a full radar volume and 2.5 min to get a half volume. Cells moving fast with respect to the radar may have been displaced significantly, resulting in a

tilted-shape cell and the cell core may already have partially or even totally left the area where the cell is estimated to reside. A final source of uncertainty is geometrical, which has two main issues. In the first place there is an issue with the minimum and maximum visible altitude by the radar. Assuming a typical melting layer top placed at 3000 m MSL and no beam blockage, the distance at which the radars in the Swiss radar network may observe the base of the RPC ranges from 200 km for the

radar placed at the lowest elevation (Albis) to on the order of 40 km for the highest placed radar (Plaine Morte). Obviously, in areas of beam blockage (the Alps and the Jura mountains), such ranges are further reduced. The maximum visible altitude close to the radar is determined by the extend of the so-called cone of silence. For example, in the case of the Albis radar, the maximum visible altitude does not reach 10000 m MSL until up to 30 km from the radar. Furthermore, in order to reduce the data size, the highest radar beams maximum range is capped. Consequently, at ranges further than 160 km, the maximum

visible altitude is reduced to below 10000 m MSL. Another source of error is related to how well the radar volume is sampled. Since the dominant hydrometeor is determined using all the data within the resolution volume formed by the cell area and the height resolution, the sampling is determined by how many radar gates cover such volume. If the cell area is small and/or there is a large gap between beams, it may happen that few or none radar range gates can be used in the sampling and therefore gaps may appear in the RPC.

Considering all the sources of error aforementioned, we estimate that for individual radars with good visibility, RPC can provide useful information in a range between roughly from 20 to 80 km. Outside of that range, they may still provide useful information but it should be considered qualitative in nature. In a dense radar network such as the Swiss one, the RPC coverage can be extended by making use of a radar composite, which benefits of the observation of the same column by multiple radars. For the purpose of this study though, we assume that, at least in the LMA coverage restricted domain discussed in the following

subsection, the RPC coverage is sufficient.

## 2.2 Ligthning mesurements

Lightning detection was performed using two networks, the European Cooperation for Lightning Detection Network (EU-CLID) Schulz et al. (2016) and an LMA network specifically deployed for the campaign. Both networks are described in more detail in Figueras i Ventura et al. (2019). It is sufficient to mention here that the EUCLID network has a high CG flash detection

efficiency (on the order of 95%), but a reduced IC detection efficiency. The EUCLID network provides information about the location of the lightning strokes, intensity and polarity. The LMA network consisted of 6 VHF sensors which were provided by LRG. The LMA sensors detect VHF sources generated by lightning leaders and provides 3D information about the source location (including the altitude). A post-processing algorithm then clusters together individual sources deemed to be part of the same flash and assigns them a unique ID number. Since the detection of the lightning leaders requires direct line of sight of

the source, LMAs observe with high efficiency intra-cloud (IC) activity, mostly from negative leaders moving through regions of positive charge. However, weaker sources from positive leaders moving through negative charge regions are often detected (van der Velde and Montanyà, 2013). In complex orography, cloud-to-ground (CG) activity is often detected indirectly from stepped negative leaders or, less often, from negative dart leaders and some times positive leaders as well.

## 3 General data analysis

The LMA was installed in the Säntis area between 29 June and 15 August 2017. For half of these days (24 out of 48) some lightning activity was registered in the LMA-covered area by the EUCLID network. During 15 of these 24 days lightning activity was registered within 2 km from the Säntis tower. Within the 15 days of interest, there were a total of 257 cells

crossing the reduced LMA domain visible in Fig. 1. On 8 of those 15 days, LMA data were available. 22, 24 and 25 of July were excluded because less than 5 LMA stations were operational. Days 5, 8, 9 and 15 August were excluded too because, even though enough stations were operating, the data quality was poor. The reason for the poor quality is still under investigation.

The large majority of cells in the analysis (211) had a maximum severity rank of weak, 23 reached a rank corresponding to developing, 15 moderate and 8 severe. None were classified as very severe. The maximum rank achieved by a cell was 3.4. In

178 cells no EUCLID CG strokes were detected. Those cells were for the most part classified as weak or developing. Only one cell reached moderate status (2.1). The number of cells during days where LMA data were analyzed was 147. Out of these, 54 cells had lightning activity according to the LMA.

Most cells were traveling from west/south-west to east/north-east. The cells that were most severe at the time when they crossed the reduced LMA domain originated from outside of it and were crossing it at an already fairly mature stage. Due to

the relatively small area covered by the reduced LMA domain, the cells that spent their entire lifetime within its boundaries tended to be shorter-lived and weaker since they either dissipated early without growing in severity or they abandoned the reduced area. The highest rank of a cell generated and dissipating within the domain was moderate (2.1).

Fig. 2 shows a scatter plot of the number of flashes detected by the LMA network versus the number of CG strokes detected by the EUCLID network within a TRT cell. Only the lightning activity of cells transiting through the reduced LMA domain

have been plotted. As it can be seen, there is a low correlation between the number of flashes detected within a TRT cell by the LMA network and the CG strokes detected by the EUCLID network. Consequently, it can be inferred that there is no linear relation between the intra-cloud and the cloud-to-ground lightning activity.

We are interested in determining whether any meaningful relationship can be established between radar data signatures and lightning activity. We will analyze radar data with respect to two metrics for lightning activity: the absolute number

of flashes/sources within the cell domain and the density of flashes/sources considering the cell area. This second metric is used in order to account for the varying dimensions of a TRT cell due to the dynamic thresholding scheme. Fig. 3 shows a scatter plot of the TRT cell rank versus the number of CG strokes detected by the EUCLID network (left panel) and the CG stroke density (right panel). The figures show that there is a very weak correlation between cell ranking and lightning activity. The situation is slightly better when confronting rank versus LMA flashes (Fig. 4) or LMA sources (Fig. 5). In these cases,

although a significant number of points with high rank have few lightning activity, there seems to be an incremental increase in the number of flashes starting from rank 1. When considering only the reduced domain, some of the high rank points with no lightning activity disappear. Perhaps a bit counter-intuitively, the absolute number of flashes/sources seems to provide better correlation. This is due to the small size of the weak to moderate TRT cells. The number of flashes has better correlation with the cell rank than the number of sources.

We now examine the time of occurrence of the first maximum of lightning activity with respect to the time of occurrence of the first maximum rank within the TRT cell (see Fig. 6). Of the 257 cells for which EUCLID data are available, only in 79 were there flashes detected within the cell. In 22% of those cells, the first maximum of strokes within the cell (19% when stroke density is considered) occurred before the maximum rank of the cell was achieved, 13% (13% as well) occurred simultaneously and 66% (68%) occurred after the maximum rank was achieved.

Of the 147 cells where LMA data was available, only 54 cells exhibited some lightning activity. Of these, in 19% of the cases the first maximum of flashes within the cell occurred before the first maximum rank was achieved (17% if flash density is used), in 24% of the cases it occurred simultaneously (28% for flash density) and 57% occurred after the first maximum rank (56% for flash density). If we consider only cells that move strictly within the LMA domain (not shown), there are 84 of these but, since they are in general short-lived and weak, only 19 of them produced lightning activity. However, the ratio is approximately the same with 16% getting a maximum of flashes before the maximum rank, 37% simultaneously and 47% after. In fact, examining the cell rank when the first maximum of lightning activity is achieved (Fig. 7), it can be observed that in general it is not particularly high, suggesting that the peaks in lightning activity occur either at the development phase of the convective cell or once the cell is mature.

The prevalence of peaks of lightning activity well after the maximum cell rank is achieved is in line with past studies that have shown that very severe cells tend to have a reduced lightning activity right before and during their mature phase (e.g., Montanyà et al., 2007). The hypothesis for that is that the strong updraft characteristic of severe cells would lift the charge centers higher up (thus making it less likely for flashes to reach the ground) and prevent particles to grow and acquire charge at a given level (thus reducing the IC flashes likelihood). Detailed studies of specific convective systems have reached similar conclusions (e.g. Lang and Rutledge, 2002; Tessendorf et al., 2005; Wiens et al., 2005).

From the data analysis in Figueras i Ventura et al. (2019), we inferred that most lightning activity was produced in areas where rimed particles or hail were predominant. Fig. 8 shows a scatter plot of RPC height versus flashes for both the EUCLID network CG strokes and the LMA network flash origins. Only data from the reduced LMA domain and from days where the LMA was active are considered. For the LMA detected flashes in particular (bottom panels), the correlation is rather high. Noticeable lightning activity starts roughly at a column height of 2000 m and largely increases with increasing height. Cloud to ground activity may occur even with modest heights, but a long column (over 8000 m) is a strong indicator of intense lightning activity. We have divided the data for CG activity (top panels) into CG+ (blue crosses), CG- (red crosses) and total lightning (green dots). Again, there is a marked increase in lightning activity with RPC height, particularly above 8000 m. It is worth noticing that when the RPC reaches such heights, an important proportion of the lightning activity is due to CG+ strokes. Moreover, when no RPC was retrieved, i.e. RPC height equals 0, the dominant type of stroke has positive polarity. If we examine the CG activity as a function of RPC base altitude (Fig. 9), it can be observed that there is a weak dependency. Indeed, lightning activity is low when the column starts above 5000 m MSL, although it must be noticed that there is a large peak in activity when the RPC base is located at 4500 m MSL. In such case a significant percentage is constituted by CG+ strokes. From this analysis, it can be inferred that RPC length is a better indicator of lightning that RPC base altitude, although

a more adequate metric would perhaps be height with respect to the average/maximum ground altitude of the area covered by the TRT cell.

## 4 Case study: Comparison of two severe TRT cells with different lightning efficiency

We now analyze in more detail two of the most severe cells encountered during the campaign. The first one occurred on day 2017-07-19 (TRT cell ID 2017071915100055 hereafter cell 1) and reached a maximum rank of 3.1, while the second occurred on day 2017-08-01 (TRT cell ID 2017080116050003 hereafter cell 2) and reached a maximum rank of 3.4. Both are therefore classified as severe. They however diverge significantly in the number of CG strokes produced. While the first cell produced a peak of 16 CG strokes (stroke density of 0.035 $\mathrm{flashes \cdot km^{-2}}$), the second one reached 130 strokes (stroke density of 0.48 $\mathrm{flashes \cdot km^{-2}}$).

Fig. 10 shows graphs of the position, velocity and area of the two cells. The duration of the cells was similar (cell 1: 2h, cell 2: 2.5h) and they were first detected at similar times (Cell 1 at 15:10 UTC, cell 2 at 16:05 UTC). Cell 1 started south, close to the Alps, and moved somewhat erratically from south-west to north-east, following the footsteps of the Alps. Initially it moved very fast towards north but then rapidly lost speed. Cell 2 moved rather fast on a narrow strip from west to east. Cell 1 started with a small area of less than $100\,\mathrm{km^2}$ and progressively grew up to more than $400\,\mathrm{km^2}$, then it likely split at 16:10 UTC and merged again with another cell at 16:35 UTC, thereby reaching the maximum area. Cell 2 started with an already large area of more than $200\,\mathrm{km^2}$, progressively grew up to $600\,\mathrm{km^2}$ and at 17:35 UTC it split into two and kept an area of roughly $200\,\mathrm{km^2}$.

In terms of ranking (see Fig. 11 top panels), cell 1 started as weak (ranking 0) but reached moderate status rather fast (15 min) and stayed in that category for most of its lifespan, except for two time steps ranked as severe in the first half of its life (15:50 and 15:55 UTC). Cell 2, on the other hand, was already developing when first detected, but it took 35 min to reach moderate status. It reached the category of severe at two time steps during the second part of its life, at 17:20 UTC and 17:25 UTC. Cell 1 had very low CG lightning activity during the first part of its life (see Fig. 11 middle panels). The few lightning strokes produced during this period all had positive polarity except for one. At 16:10 UTC, there was a first peak in lightning activity and during the last part of its life, it remained modestly active. The maximum activity was achieved at 16:40 UTC, although at this point the cell was relatively large so the stroke density was quite modest. For the most part of its lifetime, there was a higher percentage of CG+ than CG-. Interestingly, there were no CG strokes detected during its peak ranking. Cell 2 had plenty of CG lightning activity during its entire lifespan. The peak maximum of 120 CG strokes was reached early on and well before the maximum rank was reached. In fact, at that time, the cell rank was a modest 1.1 (weak). During the first part of its life, until 16:50 UTC, there were very few CG+ strokes detected. After 16:50 UTC there was another increase in ligthning activity but this time a significant proportion of strokes had positive polarity. The lightning activity remained rather high virtually till the end of the cell, with a significant proportion of CG+ strokes, although that proportion was never as high as in cell 1.

Unfortunately, the cells only crossed the domain of maximum LMA detection for part of their life span (see Fig. 11, lower panels). However, this coincided with peaks of activity in the cell (towards the end of its life for cell 1 and midlife for cell 2). Significant differences can already be seen in the number of LMA flashes detected within the cell with respect to the number of CG strokes. Whereas cell 1 had a significant number of flashes detected (between 100 and 200), producing very modest

CG activity (a maximum of 16 flashes), cell 2 had an even higher LMA flash detection (between 250 and 350) and a large number of flashes reached the ground (well above 60). When looking at the altitude where those (LMA) flashes originated and propagated in areas with good LMA detection, differences between the two cells can be seen. (Fig. 12). In cell 1, most of the flashes originated within a narrow band between roughly 7000 and 9000 m MSL. In cell 2, the origin of the flashes is more widespread but at the same time, higher concentrations can be found higher up in the atmosphere towards 9000 m MSL.

Looking at the position of all VHF sources, it is clear that the flashes of cell 2 propagated preferably at a lower altitude.

The main differences between the two cells though, can be observed in their vertical profiles (see Fig. 13, Fig. 14 and Fig. 15). Cell 1 had an RPC base altitude of roughly 4000 m MSL for its entire lifespan. Initially, the RPC height was about 4000 m and progressively grew (with some fluctuations) up to 7000 m on average, coinciding with the time of maximum CG flash activity . Cell 2, on the other hand, had a slightly higher RPC base altitude, around 4500 m MSL for most of its lifespan. What

is noticeable is that the RPC height was much higher than that of cell 1. From the beginning of its life, it was on the order of 8000 m and reached a length of more than 10000 m in the second part of its life, where it also had a significant amount of hail in it. Notice that after the cell split, and due to the reduced size of the resultant cell, there were height levels where few radar data was available, hence the data gaps. The reflectivity values were also higher in general and with values higher than 30 dBZ well past 12000 m MSL, suggesting the presence of larger and more abundant particles. $\rho_{hv}$ values above the freezing level

were lower, particularly during the second part of its life, when large amounts of hail were present, a hint about the variety of particle shapes. It is also interesting to notice the large negative values of $K_{dp}$ during the second half of its life span, a feature that has been associated with lightning activity (Ryzhkov and Zrnic, 2007; Figueras i Ventura et al., 2013). In general, cell 2 exhibited large values of $K_{dp}$ below the freezing level. Cell 1 had also larger values of $K_{dp}$ below the freezing level when there was an increase in lightning activity but one should also be cautious when interpreting this. Since the cell was moving very

close to the Alps, it is likely that the radar coverage was poor at those low altitudes. Focusing on the frequency of occurrence of each value at each time stamp for cell 2 (see Fig. 16), it is worth noticing that during the first peak of lightning activity, there was a marked shift of the histogram of both $Z_{dr}$ and $K_{dp}$ towards positive values, a feature that may indicate the presence of $Z_{dr}$ columns, i.e. large updraft (Snyder et al., 2015).

## 5   Conclusions

In this paper, we have presented an analysis of a large dataset of convective cells detected over the course of a lightning measurement campaign that took place in the summer of 2017 in the area surrounding the Säntis mountain, northeast of Switzerland. In this campaign, for the first time in the Alps, a lightning mapping array was deployed. The use of the operational

EUCLID network and the LMA network allows a thorough analysis of both the intra-cloud and the cloud-to-ground lightning activity within the convective cells.

The main conclusions of this study are:

- In general terms, an increase of IC lightning activity (as detected through the LMA) resulted in an increase of CG activity (as detected through the EUCLID network). However, there were several outliers. In one case, an excess of 500 LMA flashes resulted in few CG strokes while in another, 30 CG strokes were detected without any apparent IC activity. We can thus conclude that there is no linear relation between IC and CG lightning activity.

- Cells without lightning activity during their life cycle were indeed classified as weak. However, the rank of the convective cell is a poor indicator of its lightning activity, particularly considering CG flashes. In half of the cells studied, the maximum of lightning activity was reached after the maximum rank was reached and in a quarter it was reached before. Generally speaking, the maximum lightning activity was reached at the time period when cells were classified as weak to moderate. Our hypothesis is that this is linked to the VIL term in the ranking equation, effectively is an integral of reflectivity over height. As such, much more weight is given to the mixed-phase and liquid regions of the precipitating system which, due to the large dielectric constant of their hydrometeors, have a much larger reflectivity. However, it has repeatedly been shown in literature that increases in lightning rate tend to happen before and after the most severe (on the ground) phase of the convective precipitation.

- A more promising predictor of lightning activity seems to be the altitude of the rimed particles column, particularly for IC flashes. An increase in lightning activity was clearly shown from 3000 m onward. High CG lightning activity was observed when the rimed particle column was larger than 8000 m.

- The detailed study of two cells with similar characteristics but with different levels of CG lightning activity showed that there were significant differences in the composition of the solid phase region of the convective cloud. The cell with less lightning activity had a shallower RPC, a lower proportion of hail and in general lower reflectivity values and higher $\rho_{hv}$ values, suggesting smaller and more homogeneous particles.

This study has shown the usefulness of an LMA network even in a complex terrain such as the Swiss Alps in order to better characterize the intra-cloud lightning activity. It has also shown that a new polarimetric radar-based parameter, the rimed particles column, may be used within the context of cell severity warnings to add more explicit information about lightning activity. From this study, it can be concluded that a radar-based lightning nowcasting system should be essentially probabilistic and take into account among other things the rimed particle column height and base altitude as well as the orography and man-made structures or, alternatively, the lightning climatology.

*Code and data availability.* Code used to post-process the radar data is available on github https://github.com/meteoswiss-mdr. Data is available on request by contacting the authors.

*Author contributions.* JFV performed the radar data processing and the data analysis contained in this paper. NB and JG contributed to the radar data processing and data interpretation. OvV, DR, JM, NP, AS, AM, MA, MR and FR deployed the LMA network and processed its data. UG and AH advised on the content of the manuscript. JFV, with contributions from all authors, prepared the manuscript.

*Competing interests.* The authors declare that they have no conflict of interest.

5 *Acknowledgements.* This work was partially supported by the Swiss National Science Foundation (Project No. 200020_175594), the European Union's Horizon 2020 research and innovation program under grant agreement No 737033-LLR, research grants from the Spanish Ministry of Economy and the 421 European Regional Development Fund (FEDER): ESP2013-48032-C5-3-R, ESP2015-69909-C5-5-R422 and ESP2017-86263-C4-2-R. We thank Nathalie Caloz for her thorough proofreading.

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

**Table 1.** TRT cell parameters computed operationally.

| Parameter | Units | Definition |
|---|---|---|
| $det$ | dBZ | Reflectivity cell detection threshold |
| $RANKr$ | – | $10 \cdot RANK$ of the cell |
| $ET45$, $ET45m$ | km MSL | Maximum/Median altitude of the 45 dBZ reflectivity echo |
| $ET15$, $ET15m$ | km MSL | Maximum/Median altitude of the 15 dBZ reflectivity echo |
| $VIL$ | kg $\cdot$ m$^{-2}$ | Vertically integrated liquid |
| $maxH$, $maxHm$ | km | Height of maximum reflectivity (of the echo with maximum reflectivity within the cell/median within the cell) |
| $CG-$, $CG+$, $CG$ | – | Number of negative/positive/total cloud to ground ligthning strikes within the cell (from EUCLID network) |
| $\%CG+$ | % | Percentage of positive cloud to ground lightning strikes within the cell respect to total (from EUCLID network) |
| $velx$, $vely$ | km $\cdot$ h$^{-1}$ | Estimated cell velocity on the x (east-west) axis/y (south-north) axis |
| $Dvelx$, $Dvely$ | km $\cdot$ h$^{-1}$ | Standard deviation of the current time step x-axis/y-axis cell velocity respect to the previous time step |
| $area$ | km$^2$ | cell area |
| $cellcontourlon - lat$ | ° | latitude-longitude of delimiting points of a polygon enclosing the cell |
| $lon$, $lat$ | ° | longitude/latitude of the center of the cell |
| $ellL$, $ellS$ | km | Long/short axis of an ellipsis with equivalent area as the cell |
| $ellor$ | ° | Orientation of an ellipsis with equivalent area as the cell |

**Table 2.** Analyzed days with some of their general characteristics. The LMA domain refers to the yellow area in Fig. 1. The cells considered have a life span of at least 3 radar time steps (i.e. 15 min) and have been present within the domain for at least 3 time steps.

| Days examined | LMA | TRT cells within | TRT cells crossing | Max cell rank |
|---|---|---|---|---|
| 2017.06.29 | 6 | 14 | 8 | 1.4 |
| 2017.06.30 | 5 | 9 | 5 | 2.1 |
| 2017.07.10 | 5 | 12 | 13 | 1.4 |
| 2017.07.14 | 5 | 36 | 15 | 1.5 |
| 2017.07.18 | 5 | 7 | 5 | 2.6 |
| 2017.07.19 | 5 | 3 | 5 | 3.1 |
| 2017.07.22 | 4 | 1 | 6 | 2.7 |
| 2017.07.24 | 3 | 11 | 15 | 1.1 |
| 2017.07.25 | 3 | 21 | 17 | 1.3 |
| 2017.07.30 | 6 | 3 | 6 | 1.9 |
| 2017.08.01 | 6 | 0 | 6 | 3.4 |
| 2017.08.05 | 5 | 7 | 8 | 2.8 |
| 2017.08.08 | 5 | 5 | 4 | 0.5 |
| 2017.08.09 | 5 | 1 | 2 | 0.4 |
| 2017.08.15 | 5 | 9 | 3 | 1.8 |

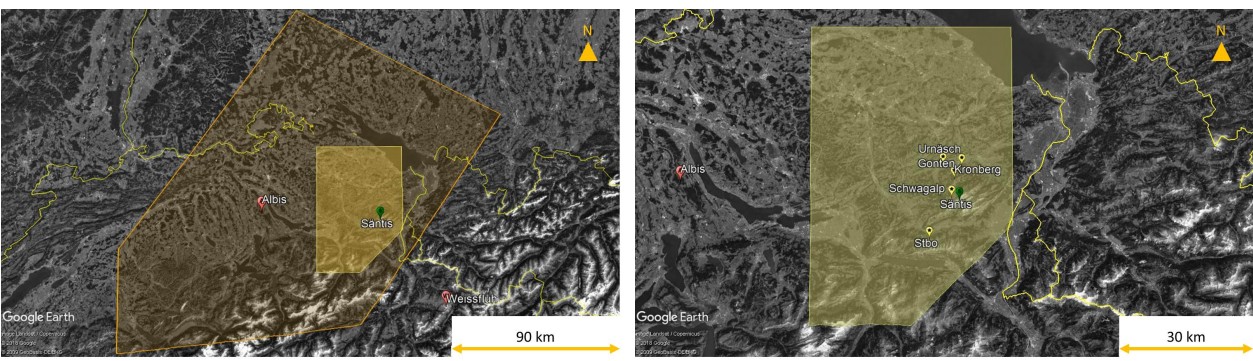

**Figure 1.** Left: Approximate extent of the maximum area covered by the LMA (Orange poligon). The yellow area shows the region with more comprehensive coverage. Right: Zoom over the best covered area with the locations of the LMA stations. Radar positions are marked by red dots while the position of the LMA sensors is marked by yellow dots. The Säntis tower is marked by a green dot.

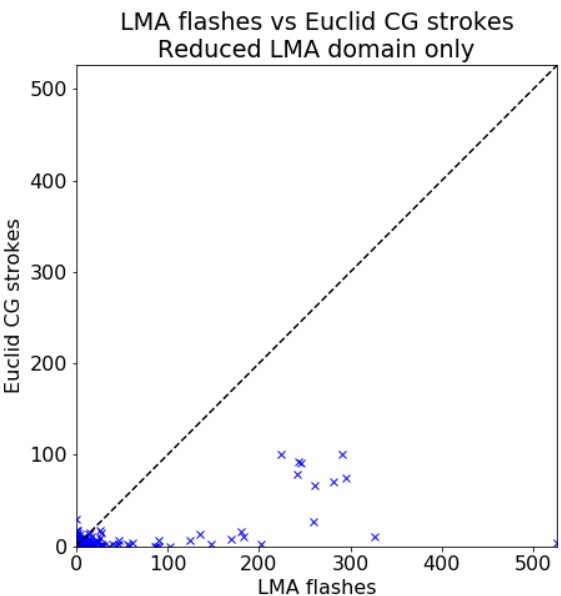

**Figure 2.** Scatter plot of the number of flashes detected by the LMA network with respect to the number of CG strokes detected by the EUCLID network within TRT-cells when transiting through the reduced domain.

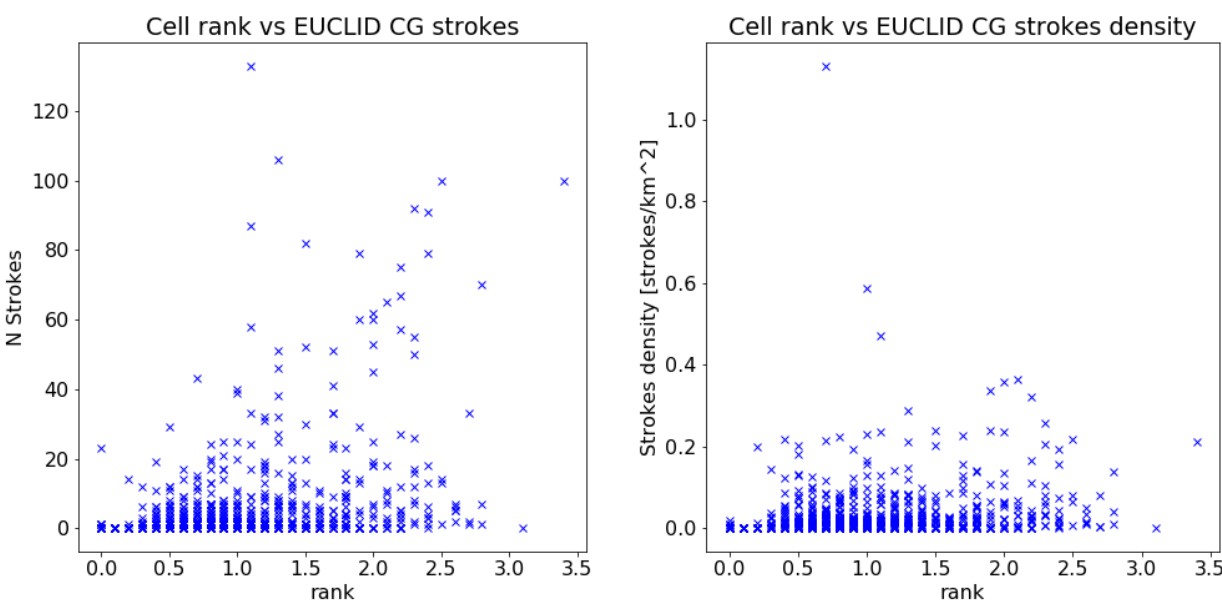

**Figure 3.** Scatter plot of rank versus: Left panel: Number of CG strokes within the TRT cell detected by the EUCLID network. Right panel: Density with respect to cell area. There are a total of 2792 points in each graph.

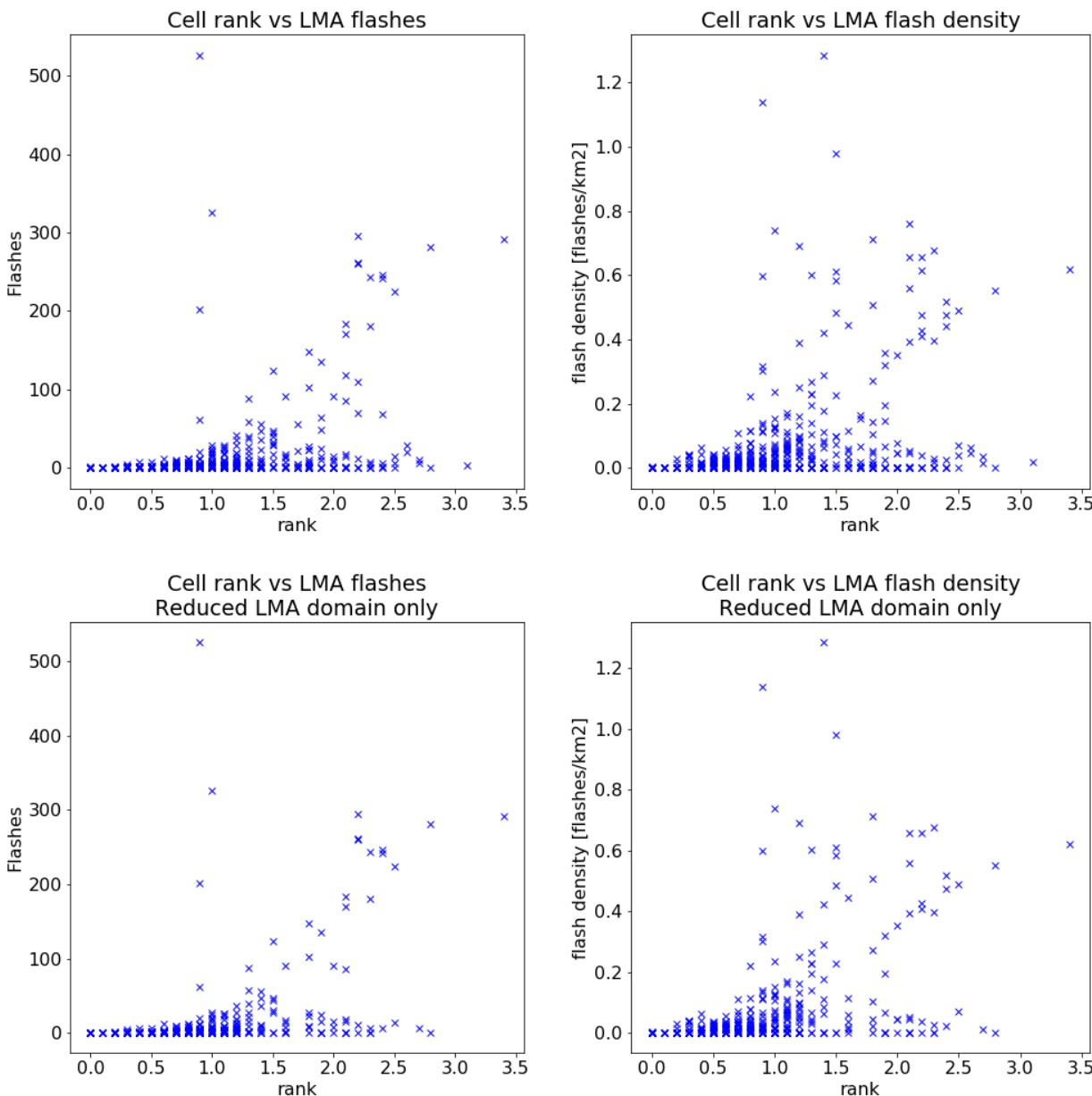

**Figure 4.** Scatter plot of rank versus: Left: Number of first VHF sources within the TRT cell detected by the LMA network. Right: Density with respect to cell area. Top: All detections, 1571 points. Bottom: Detections only when center of TRT cell within reduced domain, 965 points.

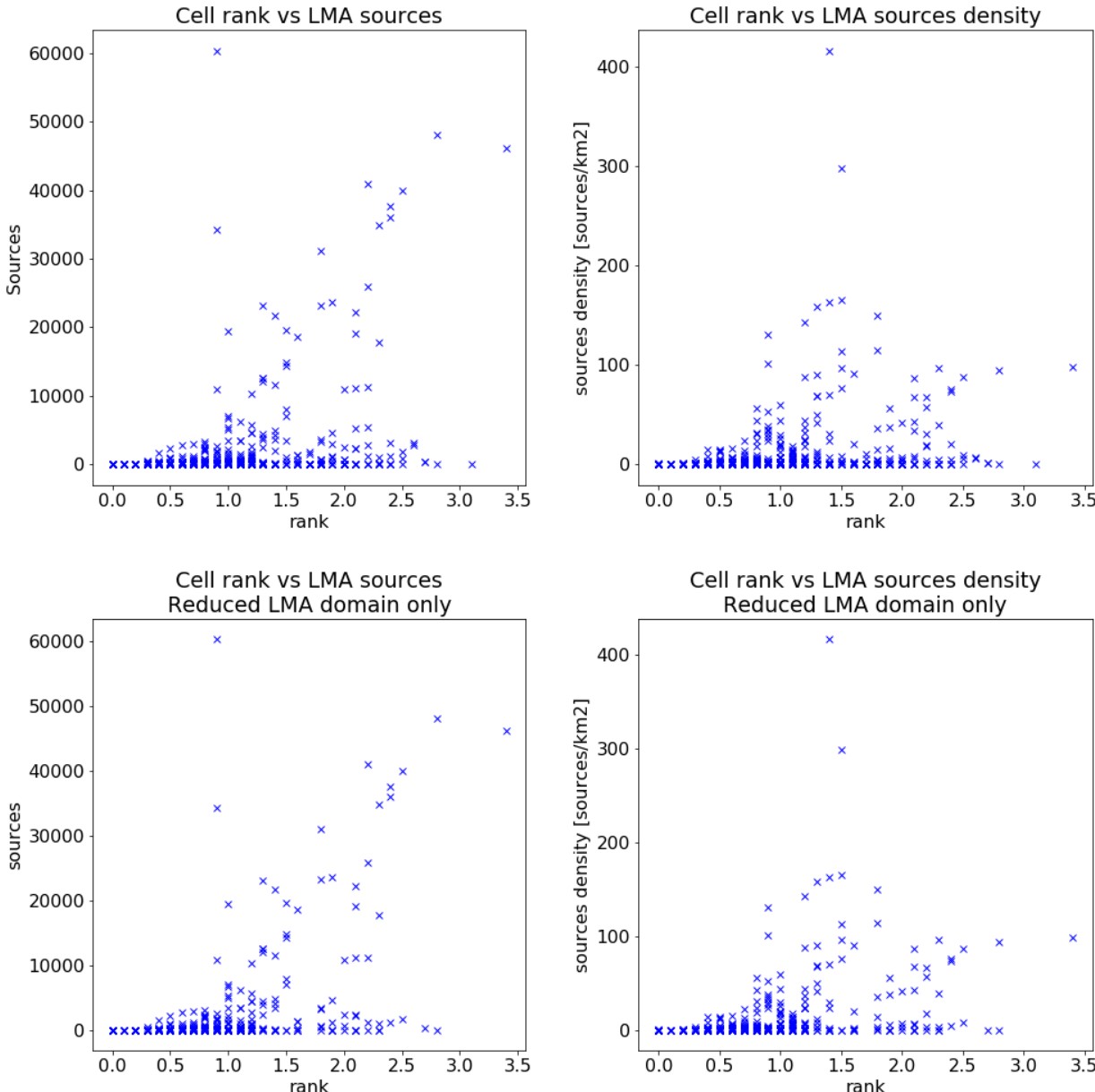

**Figure 5.** As in Fig. 4 but for LMA sources.

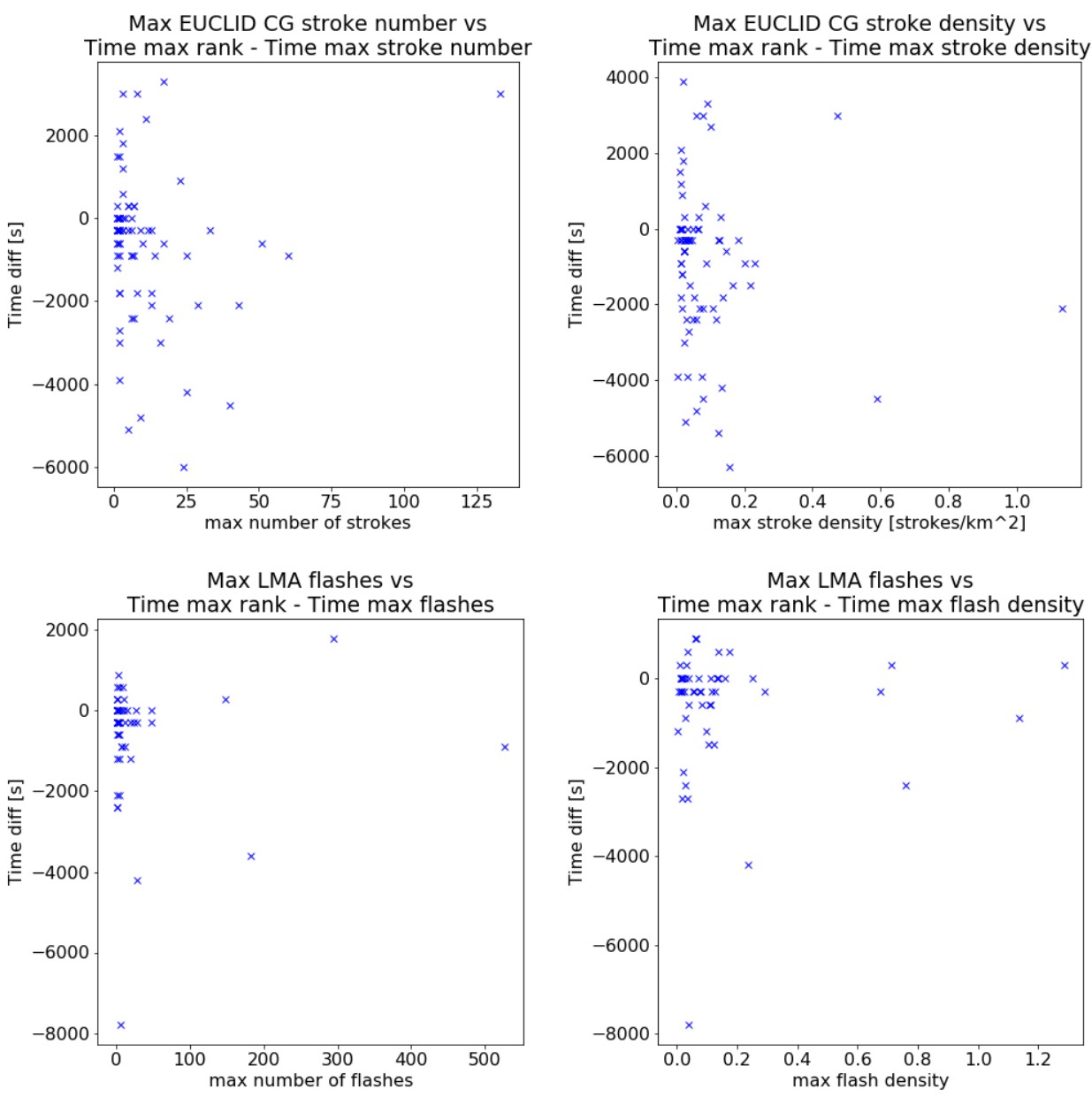

**Figure 6.** Scatter plot of: Top left: Max number of CG strokes detected by the EUCLID network with respect to time difference between occurrence of max rank and occurrence of max flashes, Top right: same as top left but for stroke density, Bottom left: Same as top left but for LMA detected flashes, Bottom right: Same as top right but for LMA detected flashes.

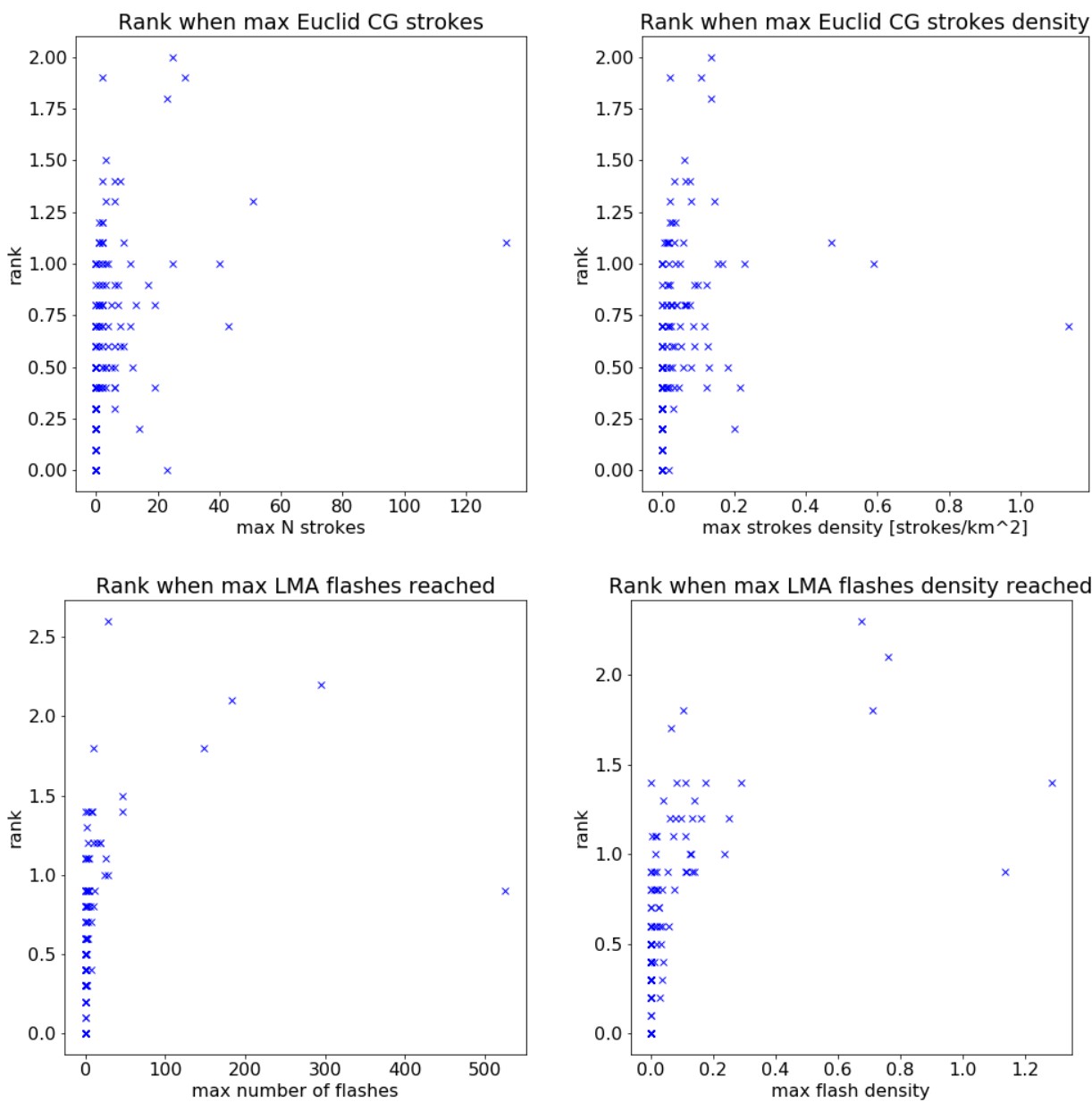

**Figure 7.** Scatter plot of: Top left: Cell rank when the first max number of CG strokes detected by the EUCLID network is achieved. Top right: same as top left but for stroke density. Bottom left: Same as top left but for LMA detected flashes. Bottom right: Same as top right but for LMA detected flashes.

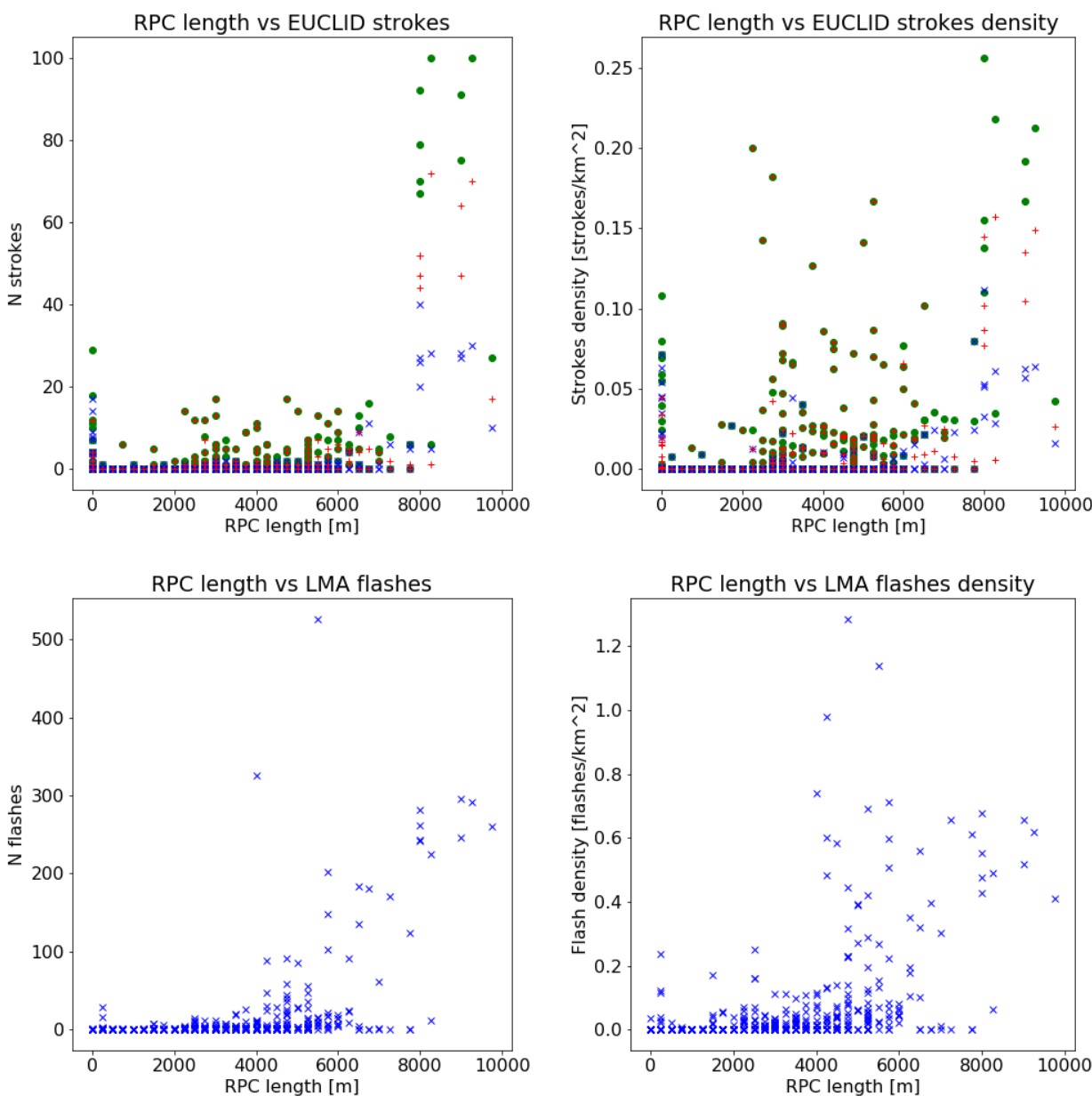

**Figure 8.** Scatter plot of RPC height versus: Left: Number of flashes within the TRT cell detected. Right: Density with respect to cell area. Top: EUCLID CG strokes. Blue crosses: CG+ strokes, red crosses: CG- strokes, green dots: total lightning. Bottom: LMA first sources.

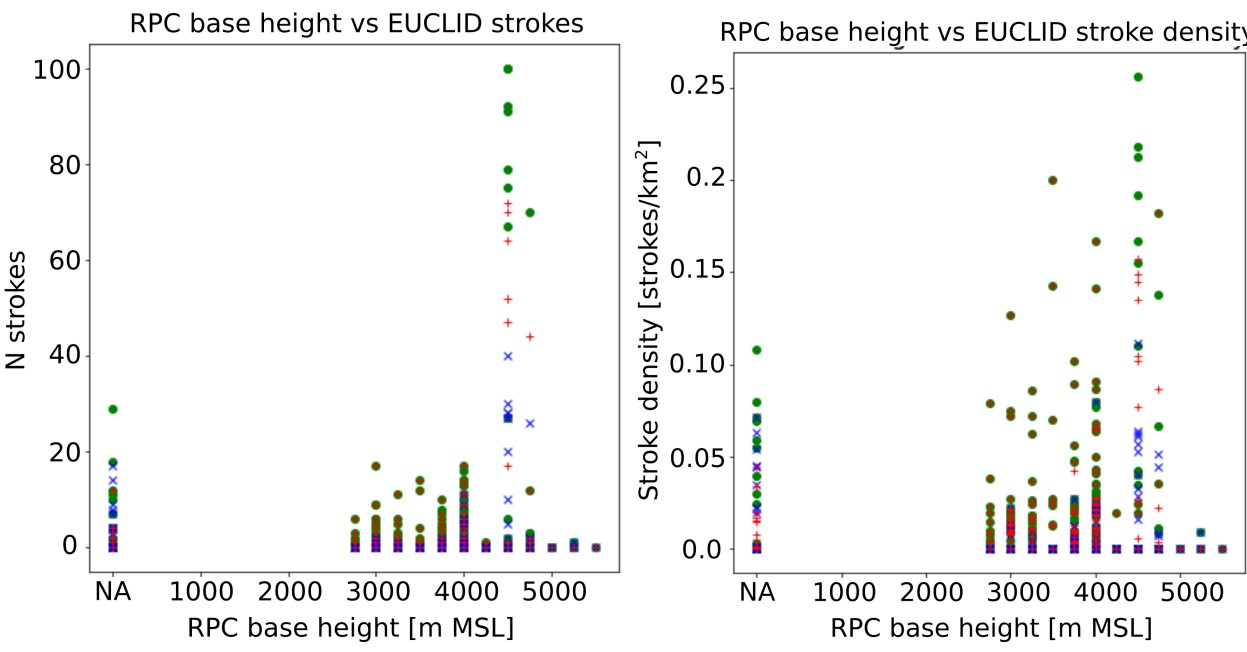

**Figure 9.** Scatter plot of RPC height versus: Left: Number of EUCLID CG strokes within the TRT cell detected. Right: Density with respect to cell area. Blue crosses: CG+ strokes, red crosses: CG- strokes, green dots: total lightning. NA means RPC base height not available.

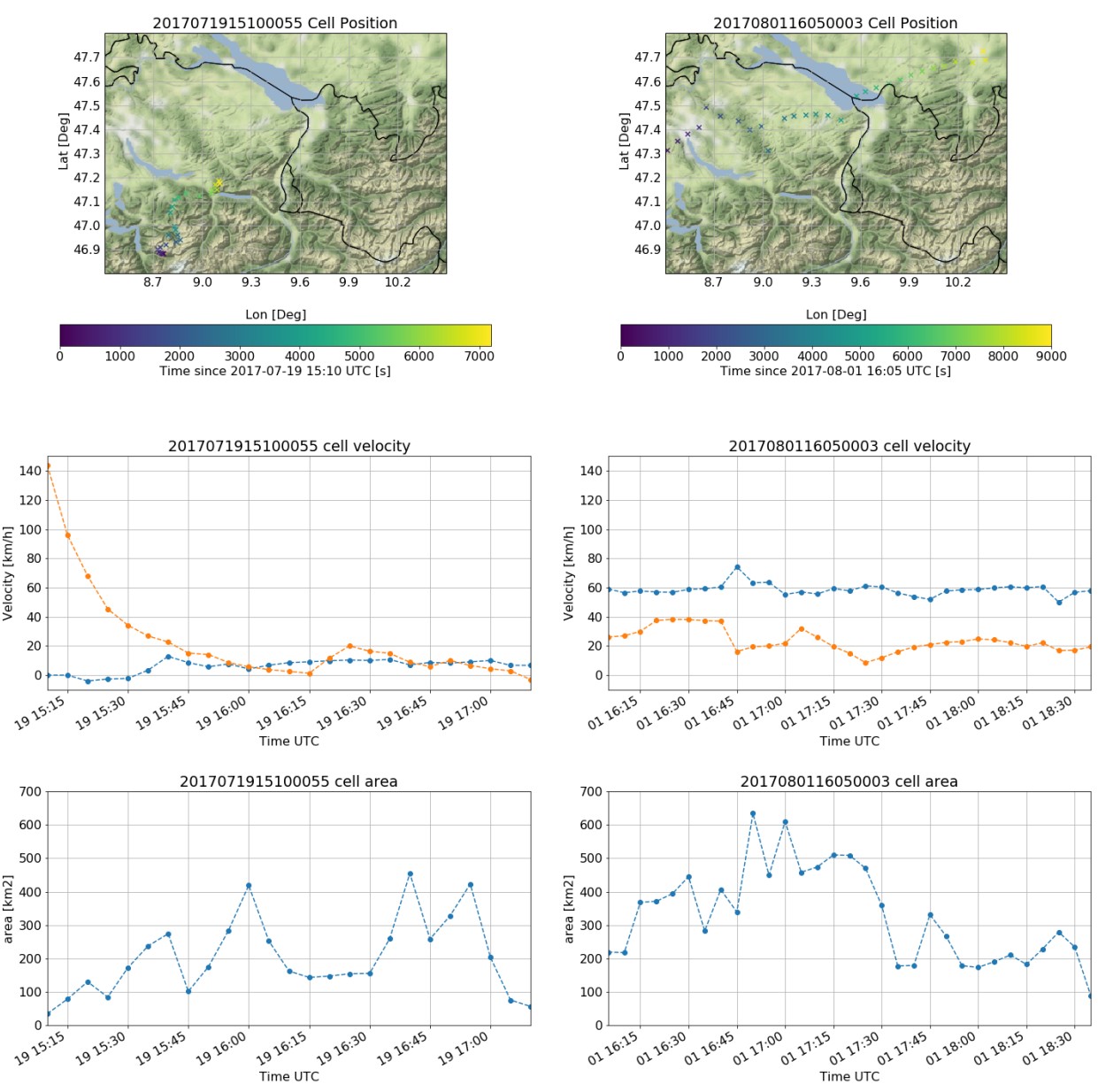

**Figure 10.** Top to bottom: Cell position, velocity (orange: South-North direction, blue: East-West direction) and area. Left: Cell 1, Right: Cell 2.

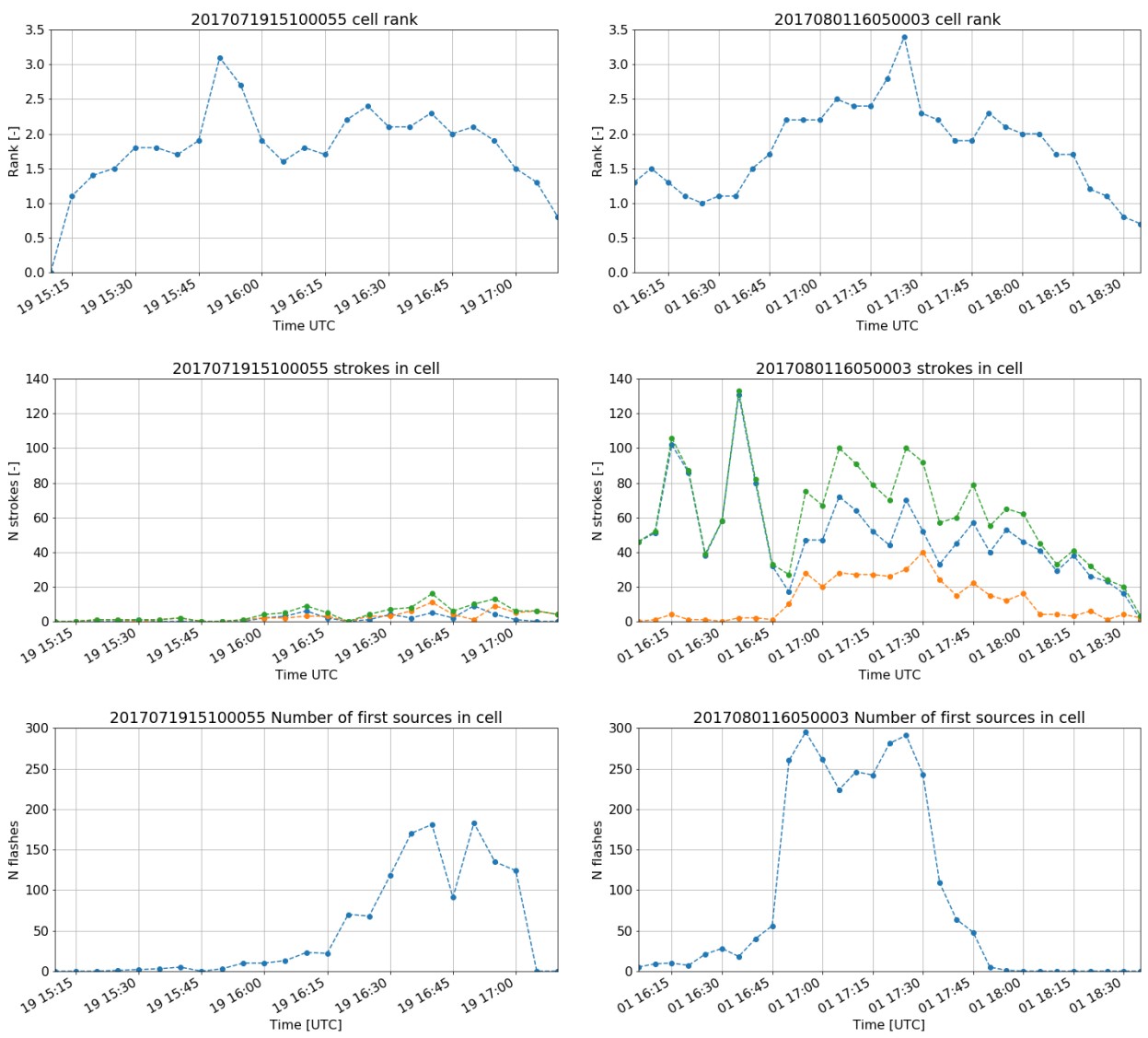

**Figure 11.** Top to bottom: Cell rank, EUCLID CG strokes (orange: positive, blue: negative, green: total) and LMA detected flashes. Left: Cell 1, Right: Cell 2.

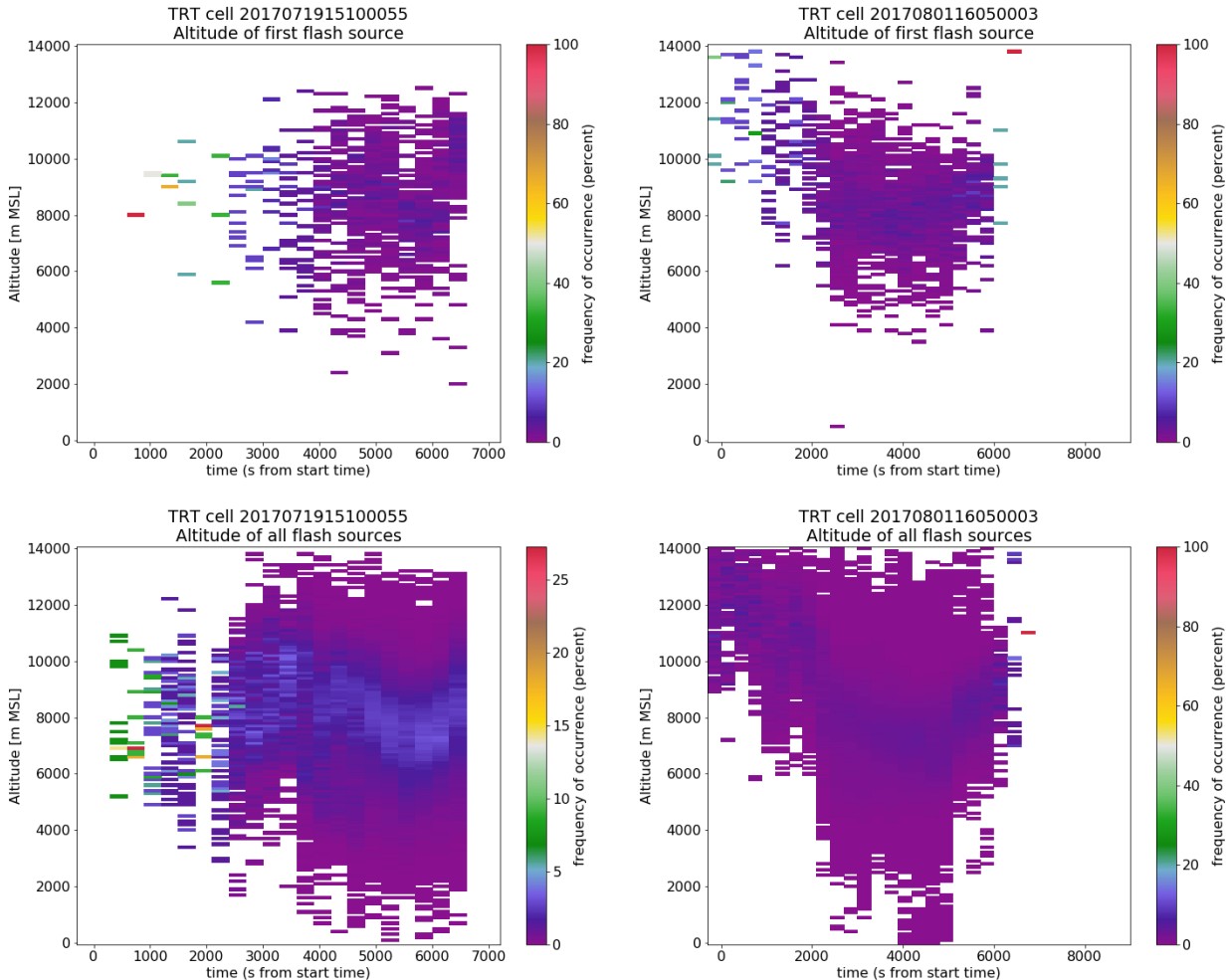

**Figure 12.** Top: Percentage of LMA flashes within the cell originated at each altitude. Bottom: Percentage of LMA flashes located at each altitude within the cell. Left: Cell 1, Right: Cell 2. Note that the percentage is computed over the total number of flashes/sources at each time step, i.e. if only one flash was detected over the time step, the corresponding height where the flash was detected will have a value of 100%.

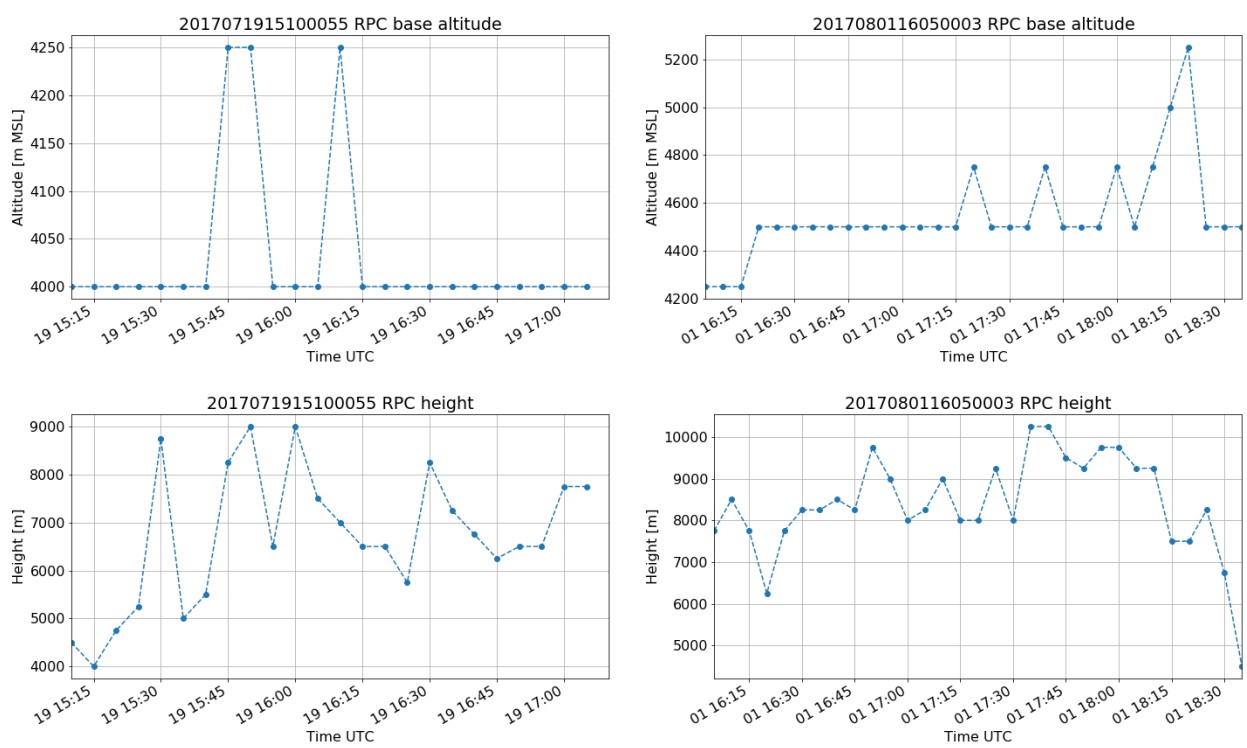

**Figure 13.** Top: RPC base altitude, Bottom: RPC height. Left: Cell 1, Right: Cell 2.

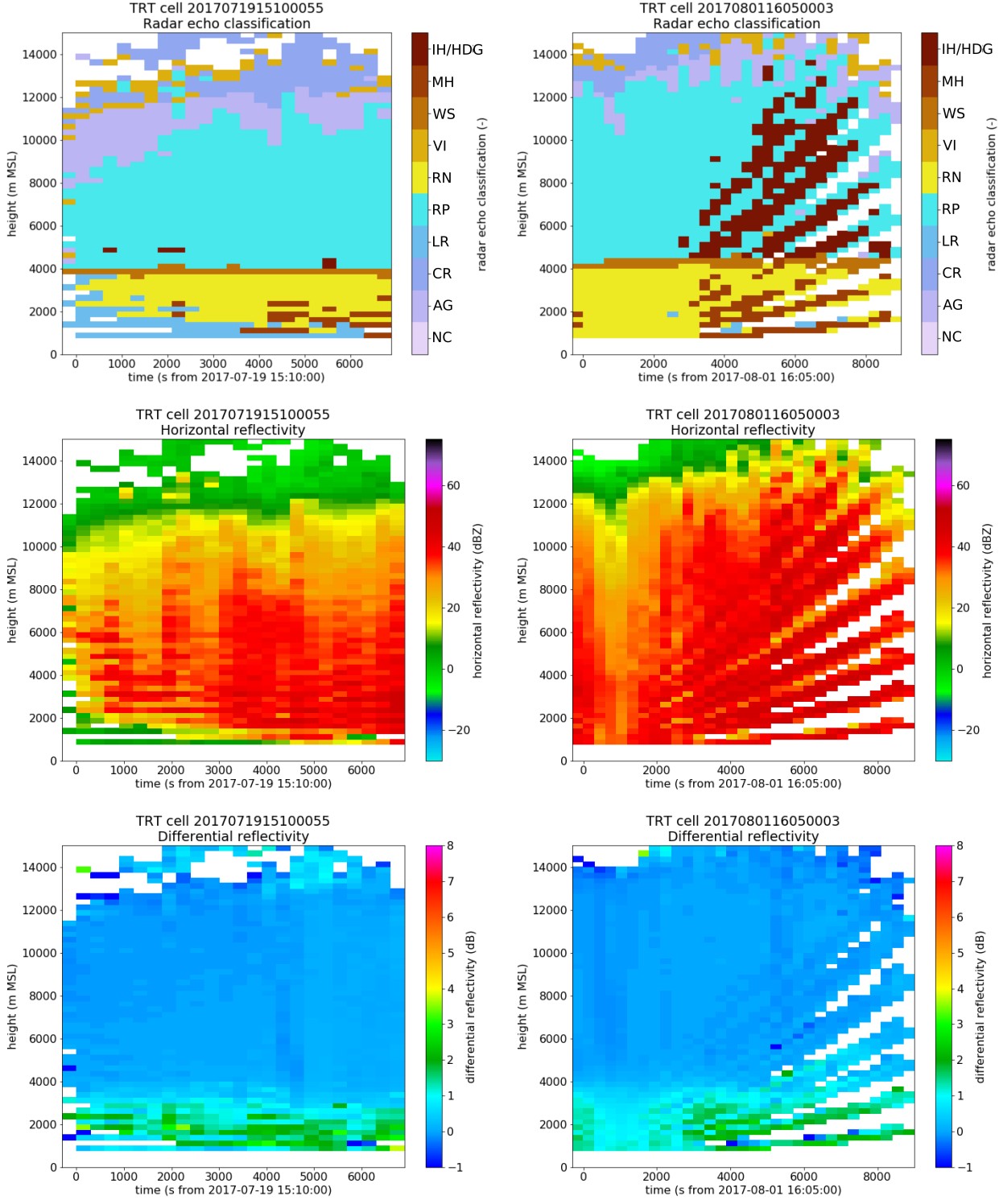

**Figure 14.** From top to bottom: Profile of hydrometeor class (mode at each level), profile of reflectivity (median values at each height level), profile of $Z_{dr}$. Left: Cell 1, Right: Cell 2

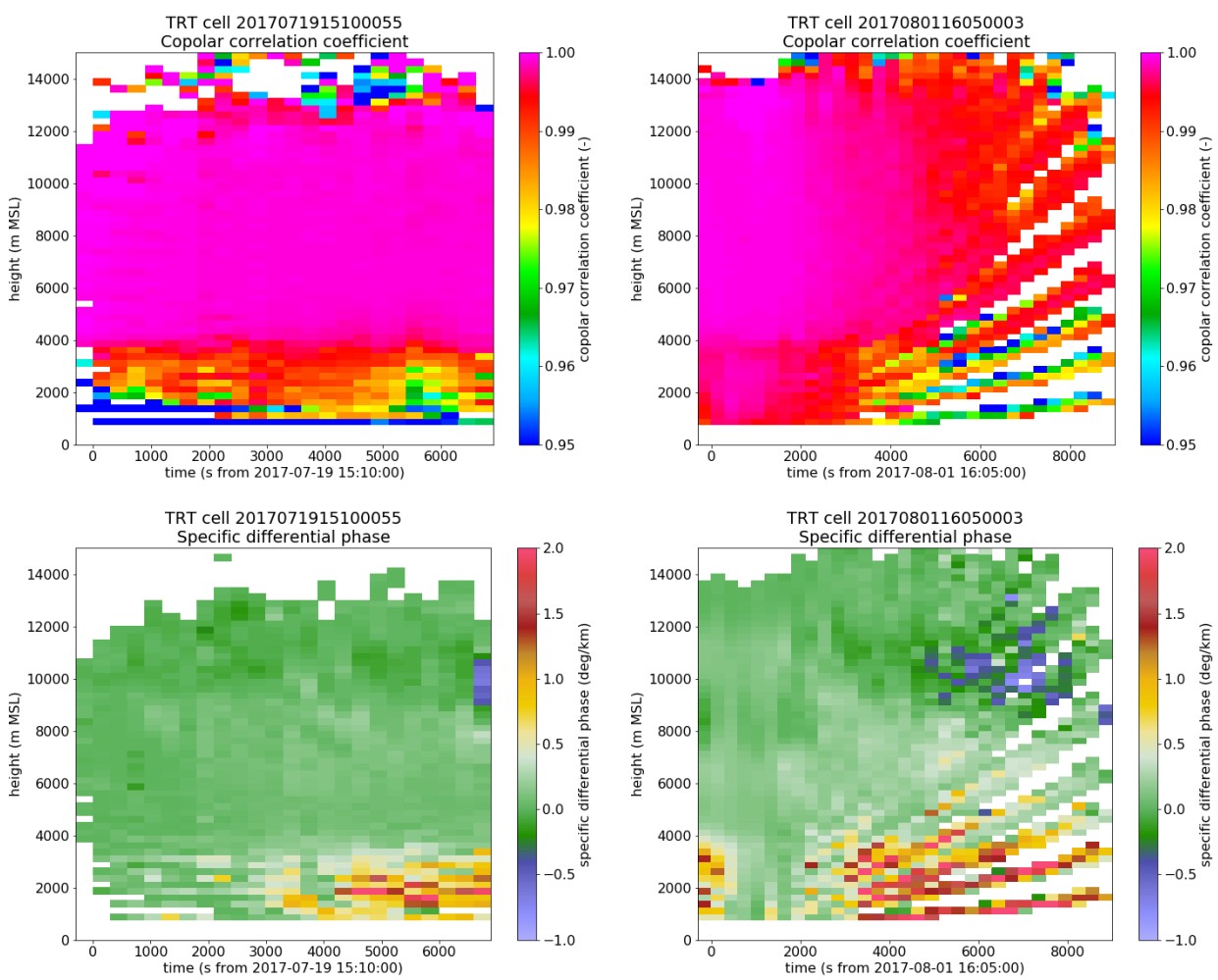

**Figure 15.** Top: Profile of $\rho_{hv}$, Bottom: Profile of $K_{dp}$. Left: Cell 1, Right: Cell 2.

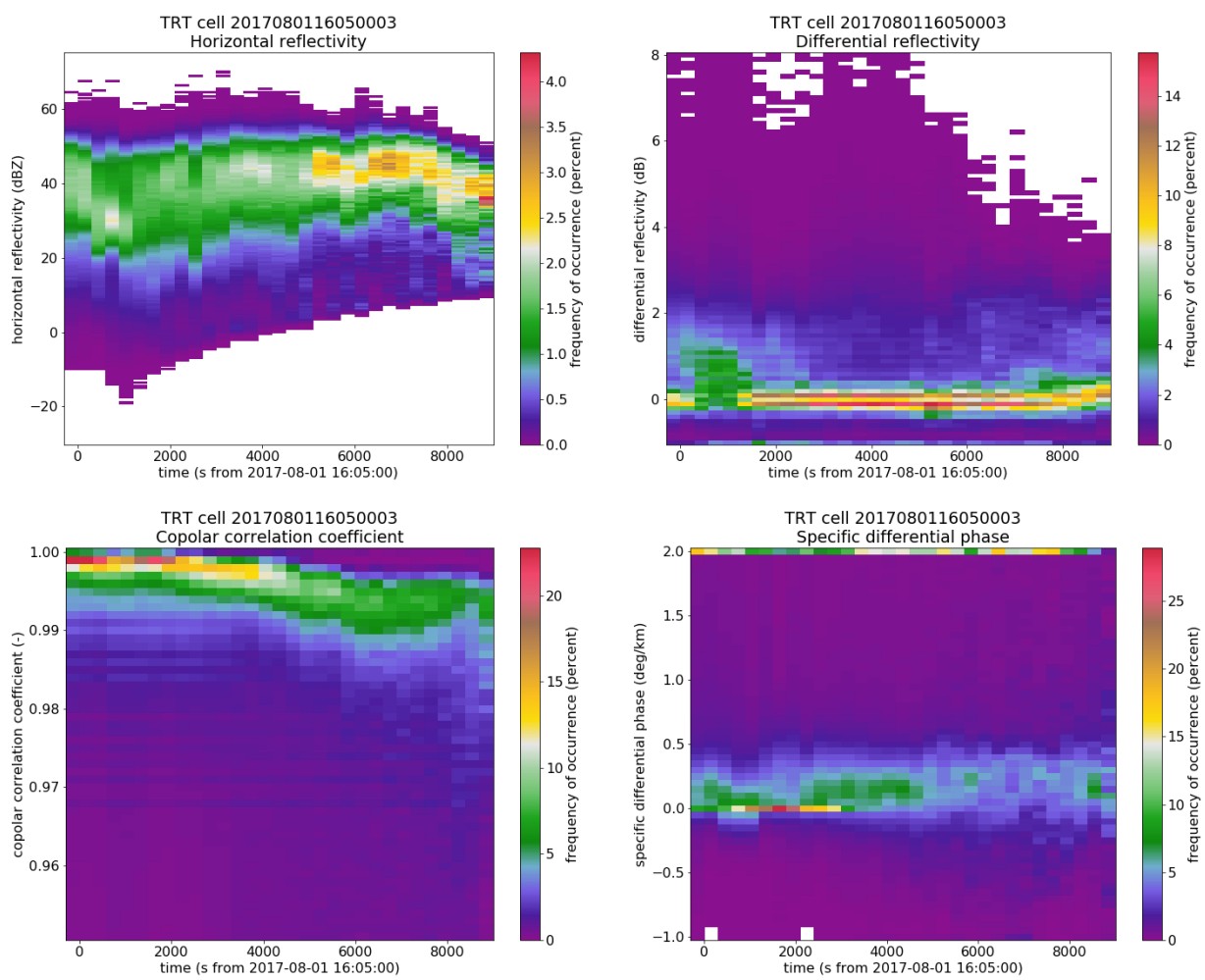

**Figure 16.** Percentage of occurrence at each time step of from top to bottom and left to right: Reflectivity, $Z_{dr}$, $\rho_{hv}$ and $K_{dp}$ for cell 2.