# Peer review of "Analysis of the lightning production of convective cells"

_Atmospheric Measurement Techniques, 2019_

## Referee Comment (RC1) · Anonymous Referee #1 · 3 Jun 2019

The paper "Analysis of the lightning production of convective cells", by Figueras i Ventura and colleagues, presents a radar-based study on the lightning production capabilities of convective cells occurred in summer 2017 in Switzerland. Weather C-band, Doppler, polarimetric radar reflectivity data are processed to automatically detect and track convective development, while lightning data were recorded from the EUCLID network, and from a Lightning Mapping Array VHF network, deployed for this campaign. The main results is that the altitude of rimed particles column is a promising predictor of lightning activity in convective cells, especially for Intra Cloud flashes: cells with less lightning activity had a shallower column, a lower proportion of hail and in general lower reflectivity values and higher values of co-polar correlation coefficient, indicating smaller and more homogeneous particles.

[Figure]

The paper is well written and addresses an important topic, contributing with new data and experiments. I think the paper should be published on AMT, after the minor corrections I suggest below.

Introduction. I think the literature review does not consider relevant papers that analysed convective systems lighting activities and cloud microphysical structure: e.g. Emersic et al., 2011, Mon. Wea. Rev., 139, 1809–1825, Wapler, 2017), Atmospheric Research 193, 60-72; Marra et al, 2017, Atmospheric Research, 192, pp. 72-90; among others.

Pag. 6. Lines 9-10. How are the number of EUCLID and LMA computed for each cell? In should be done when the cells are in the reduced domain. This figure should be discussed with more details.

Pag. 8 line 28 and following. I do not understand how figures 14 and 15 are drown. In abscissa it is time, but looking at the pictures especially for cell 2, it seems that PPI beams show up in the right part of the figure. What does it mean that (for cell 2) at 8000 s the cloud has layers with no hydrometeors? I general I suggest to better comment these figures, and to use labels to better mention them in the text.

Conclusions. The first item is not a conclusion: Authors just say that the two systems measures different things in different places. A careful detection efficiency study of the two networks should be made before the intercomparison.

––––––––––––––––––––––––––––––––––

---

## Referee Comment (RC2) · Anonymous Referee #2 · 7 Jun 2019

On 2017 during the Santis campaign, in Swiss, the unique chance for high resolution observations of convective cells and associated lightnings took place. For the first time in the Alps a LMA was set up allowing detection of intra-cloud (IC) and cloud-to-ground (CG) lightings in complex horography. The lightnings observations by LMA collected during the aforementioned campaign are analysed with EUCLID lightnings data and TRT severity rank. A general agreement between EUCLID and LMA observations is found but some relevant disagreements occurred.

TRT severity rank seems to be poorly correlated with lighting activity, while rimed particle columns is a better predictor of lighting activity. Related to their lightning activity, the detailed study of two convective cells confirmed different hydrometeor compositions between the two cells.

[Figure]

The paper is valuable, investigating one of the most relevant and still partially not well-understood process in clouds: the lighting activity in convective atmosphere. Nevertheless, the paper is largely unclear and improvements are needed to increase the paper clearness.

First of all, the language is sloppy with a lot of mis-spellings and unclear sentences (e.g. page 7 line 5 "The hypothesis for that is that. . .", or same page line 17 "Moreover, when no RPC was retrieved, i.e. RPC height equals 0, The dominant type") : a deep language review is mandatory.

General comments and recommendations. As anticipated by the generic paper's title "Analysis of lighting production of convective cells", the main focus of the paper is unclear: is it to find good predictors of lighting activity? Or, is it to relate lighting activity and TRT severity rank? Or, is it to compare EUCLID and LMA observing systems? Or, is it to assess lighting efficiency? Or all these points?

The introduction focuses on warnings based on lightning activity while it lacks of several references on lighting processes in clouds (Carey and Rutledge, 2000, Baker et al. 1995, Buiat et al. 2016, Adirosi et al. 2016 Mattos et al, 2016, Lund et al. 2008).

The analysis is quite shallow (the physical explanations of results are often missing), resulting quite confusing and hard to follow. For example at page 5 line 7 "The cells that spent their entire lifetime within the LMA domain boundaries tended to be shorter-lived and weaker. The highest rank of a cell generated and dissipating within the domain was moderate (2.1)": why does LMA area show weaker storms?

At page 7 line 4 "The hypothesis for that is that the strong updraft characteristic of severe cells would lift the charge centers higher up (thus making it less likely for flashes to reach the ground) and prevent particles to grow and acquire charge at a given level (thus reducing the IC flashes likelihood)". Here, the authors move from TRT severity rank to cell updraft. One difficulty is the relationship between TRT severity rank and updraft strength (not straightforward, indeed). Moreover, updraft–total lightning relationships of individual thunderstorms have been explored in several previous publications (Lang and Rutledge, 2002; Tessendorfet al., 2005; Wiens et al., 2005): authors

hypothesis should be framed in this contest that demonstrated the role of updraft area. The role of RPC columns is underlined by the authors. Several characteristic RPC heights derived by weather radar observations, are reported: heights error and uncertainties should be discussed and reported.

In the cells analysis time are expressed in local time?

The authors find very weak correlation between TRT severity rank and lighting activity: it is not clear why a correlation was expected. Moreover, is it the sampling statistically meaningful (only eight severe cells)?

At page 9 line 20 "Cells without lightning activity during their life cycle were mostly classified as weak but the rank of the convective cell is a poor indicator of its lightning activity, particularly considering CG flashes." The authors should try to explain the reason for this result.

The authors conclude that different ice distribution within clouds is responsible for different lighting efficiency: however, this conclusion is supported by the analysis of only two cells. It is a clue, but for a reliable assessment more cases need to be studied.

Specific comments Page 9 line 32 It is not evident why LMA network is usefulness in complex terrain: could the same result be obtained by satellite observation?

Page 10 lines 1-4 Mosier et al 2010 and Seroka et al 2012 should be taken in account as pioneering works on this topic.

Figure 1 Quite unclear: please change basemap, add distance reference, North arrow and more contrasting colors.

Figure 2 please y-axis range equal to x-axis. 500 LMA observations sound very strange. Figure 9 RPC base height equal zero m ASL is quite confusing.

Figure 10 please insert a base map with horography in track plots. Please, make y-axis with same range.

Figure 11 It is not clear the need to multiply by ten the cell rank. Please, make y-axis with same range

Figure 12 The maximum values seem to be spikes (anomaly and isolated very high values)

Figure 13 which is the RPC base or top altitude estimations uncertainties? 50 meters?

[Figure]

---

## Author Comment (AC1) · 11 Jul 2019

**Response to reviewer 1 of amt-2019-135**

In this document we provide answers to the comments of reviewer 1 of the paper amt-2019-135. Our answers to the reviewer are given in *italic* font. Proposed changes to the manuscript are highlighted in blue color.

The paper "Analysis of the lightning production of convective cells", by Figueras i Ventura and colleagues, presents a radar-based study on the lightning production capabilities of convective cells occurred in summer 2017 in Switzerland. Weather C-band, Doppler, polarimetric radar reflectivity data are processed to automatically detect and track convective development, while lightning data were recorded from the EUCLID network, and from a Lightning Mapping Array VHF network, deployed for this campaign. The main results is that the altitude of rimed particles column is a promising predictor of lightning activity in convective cells, especially for Intra Cloud flashes: cells with less lightning activity had a shallower column, a lower proportion of hail and in general lower reflectivity values and higher values of co-polar correlation coefficient, indicating smaller and more homogeneous particles.

The paper is well written and addresses an important topic, contributing with new data and experiments. I think the paper should be published on AMT, after the minor corrections I suggest below.

*We thank the reviewer for his/her positive review*

Introduction. I think the literature review does not consider relevant papers that analysed convective systems lighting activities and cloud microphysical structure: e.g. Emersic et al., 2011, Mon. Wea. Rev., 139, 1809–1825, Wapler, 2017, Atmospheric Research 193, 60-72; Marra et al, 2017, Atmospheric Research, 192, pp. 72-90; among others.

*The link between microphysical structure and lightning was discussed in a previous paper by the authors. We have now explicitly referenced that paper and added the suggested references:*

[revised manuscript text omitted]

Pag. 6. Lines 9-10. How are the number of EUCLID and LMA computed for each cell? In should be done when the cells are in the reduced domain. This figure should be discussed with more details.

*For clarification we have added the following text to section 2.1.3 describing how the number of LMA flashes per cell are computed:*

A similar rationale is used to extract data obtained by the LMA within the TRT cell footprint. Any LMA-detected lightning is assigned to the cell if it has sources located within the TRT cell area (regardless of its origin) within the time resolution of the TRT algorithm (i.e. in the last 5 min from the current TRT time stamp). Out of the resultant data, products such as the total number of flashes, total number of sources and vertical profile of the number of flashes and sources can be computed.

*We also have added the following text to section 2.1.2 regarding how the number of EUCLID lightning strokes per cell are computed:*

Among them, the number of lightning strokes is computed by counting all the strokes detected by the EUCLID network within the area covered by the cell and within the time resolution of the TRT cell (i.e. within the 5 min. prior to the current time stamp).

*We also modified the highlighted lines as follows:*

Fig. 2 shows a scatter plot of the number of flashes detected by the LMA network versus the number of CG strokes detected by the EUCLID network within a TRT cell. Only the lightning activity of cells transiting through the reduced LMA domain have been plotted. As it can be seen, there is a low correlation between the number of flashes detected within a TRT cell by the LMA network and the CG strokes detected by the EUCLID network. Consequently, it can be inferred that there is no linear relation between the intra-cloud and the cloud-to-ground lightning activity.

*And we have also modified the caption of Fig.2 :*

Scatter plot of the number of flashes detected by the LMA network with respect to the number of CG strokes detected by the EUCLID network within TRT-cells when transiting through the reduced domain.

Pag. 8 line 28 and following. I do not understand how figures 14 and 15 are drown. In abscissa it is time, but looking at the pictures especially for cell 2, it seems that PPI beams show up in the right part of the figure. What does it mean that (for cell 2) at 8000s the cloud has layers with no hydrometeors? I general I suggest to better comment these figures, and to use labels to better mention them in the text.

*We have added the following text to section 2.1.3. regarding the generation of these plots:*

More specifically, the vertical profile is constructed by taking statistics of all the valid data at a particular height interval. In the cases shown in this study, the height resolution was set to 250 m, e.g. all data at altitudes ranging from 0 to 250 m were used to compute statistics valid for that height interval. Notice that depending on the size of the cell and its position with respect to the radar, there may be height intervals where no data is present simply because no radar beam covers the sampled volume.

Conclusions. The first item is not a conclusion: Authors just say that the two systems measures different things in different places. A careful detection efficiency study of the two networks should be made before the intercomparison.

*In this item we wanted to highlight that while in general terms the intra-cloud lightning activity (i.e. mostly LMA detected) and the cloud-to-ground lightning activity (i.e. detected using the EUCLID network) are related, it is not a linear relation and there are instances where a high IC activity does not imply high CG activity and vice-versa. We have re-formulated the item as follows:*

In general terms, an increase of IC lightning activity (as detected through the LMA) resulted in an increase of CG activity (as detected through the EUCLID network). However, there were several outliers. In one case, an excess of 500 LMA flashes resulted in few CG strokes while in another, 30 CG strokes were detected without any apparent IC activity. We can thus conclude that there is no linear relation between IC and CG lightning activity.

---

## Author Comment (AC2) · 11 Jul 2019

**Response to reviewer 2 of amt-2019-135**

In this document we provide answers to the comments of reviewer 2 of the paper amt-2019-135. Our answers to the reviewer are given in *italic* font. Proposed changes to the manuscript are highlighted in blue color.

On 2017 during the Santis campaign, in Swiss, the unique chance for high resolution observations of convective cells and associated lightnings took place. For the first time in the Alps a LMA was set up allowing detection of intra-cloud (IC) and cloud-to-ground (CG) lightings in complex horography. The lightnings observations by LMA collected during the aforementioned campaign are analysed with EUCLID lightnings data and TRT severity rank. A general agreement between EUCLID and LMA observations is found but some relevant disagreements occurred.

TRT severity rank seems to be poorly correlated with lighting activity, while rimed particle columns is a better predictor of lighting activity. Related to their lightning activity, the detailed study of two convective cells confirmed different hydrometeor compositions between the two cells.

The paper is valuable, investigating one of the most relevant and still partially not well understood process in clouds: the lighting activity in convective atmosphere. Nevertheless, the paper is largely unclear and improvements are needed to increase the paper clearness.

First of all, the language is sloppy with a lot of mis-spellings and unclear sentences (e.g. page 7 line 5 "The hypothesis for that is that: : :", or same page line 17 "Moreover, when no RPC was retrieved, i.e. RPC height equals 0, The dominant type") : a deep language review is mandatory.

*We thank the reviewer for positively valuing the paper. We have thoroughly reviewed its language*

**General comments and recommendations**

As anticipated by the generic paper's title "Analysis of lighting production of convective cells", the main focus of the paper is unclear: is it to find good predictors of lighting activity? Or, is it to relate lighting activity and TRT severity rank? Or, is it to compare EUCLID and LMA observing systems? Or, is it to assess lighting efficiency? Or all these points?

*We respectfully disagree with this comment according to which the main focus of the paper is unclear. The objectives of the paper are stated quite clearly in the introduction (lines 25-30 on page 2). In short there is a need for more targeted operational warnings regarding lightning activity (both cloud-to-ground and intra-cloud). Currently, the one product at MeteoSwiss that is used to issue warnings of convective situations (where most lightning is produced) is the TRT algorithm. It is just natural that we want to investigate whether there is a strong relationship between the TRT rank and the lightning activity. The paper demonstrates that there is a weak link between the two and hence a new predictor, the RPC column, is proposed. Its performance regarding both intra-cloud and cloud-to-ground activity is assessed, both statistically and through a case study, demonstrating positive results. LMA and EUCLID data are simply used because of their different observing capabilities to better differentiate intra-cloud from cloud to ground activity.*

The introduction focuses on warnings based on lightning activity while it lacks of several references on lighting processes in clouds (Carey and Rutledge, 2000, Baker et al. 1995, Buiat et al. 2016, Adirosi et al. 2016 Mattos et al, 2016, Lund et al. 2008).

*As mentioned in the response to reviewer 1, the link between microphysics and lightning activity was discussed in a previous paper by the authors. We now explicitly mention this paper in the introduction and have also added several of the suggested references in it. See the response to reviewer 1 for the introduced text.*

The analysis is quite shallow (the physical explanations of results are often missing), resulting quite confusing and hard to follow. For example at page 5 line 7 "The cells that spent their entire lifetime within the LMA domain boundaries tended to be shorter-lived and weaker. The highest rank of a cell generated and dissipating within the domain was moderate (2.1)": why does LMA area show weaker storms?

*We have re-formulated the mentioned lines to clarify what we meant:*

Most cells were traveling from west/south-west to east/north-east. The cells that were most severe at the time when they crossed the reduced LMA domain originated from outside of it and were crossing it at an already fairly mature stage. Due to the relatively small area covered by the reduced LMA domain, the cells that spent their entire lifetime within its boundaries tended to be shorter-lived and weaker since they either dissipated early without growing in severity or they abandoned the reduced area. The highest rank of a cell generated and dissipating within the domain was moderate (2.1).

At page 7 line 4 "The hypothesis for that is that the strong updraft characteristic of severe cells would lift the charge centers higher up (thus making it less likely for flashes to reach the ground) and prevent particles to grow and acquire charge at a given level (thus reducing the IC flashes likelihood)". Here, the authors move from TRT severity rank to cell updraft. One difficulty is the relationship between TRT severity rank and updraft strength (not straightforward, indeed). Moreover, updraft–total lightning relationships of individual thunderstorms have been explored in several previous publications (Lang and Rutledge, 2002; Tessendorfet al., 2005; Wiens et al., 2005): authors hypothesis should be framed in this context that demonstrated the role of updraft area.

*We agree that the relationship between TRT severity rank and cell updraft is not so clear cut. However, our observations are in line with multiple reports in literature. We have added the suggested publications to the already mentioned article from Montanyà et al. 2007.*

The role of RPC columns is underlined by the authors. Several characteristic RPC heights derived by weather radar observations, are reported: heights error and uncertainties should be discussed and reported.

*We decided to expand the discussion of the RPC column and devote a subsection to it. We added the following text:*

2.1.4 Rimed particles column computation

[revised manuscript text omitted]

In the cells analysis time are expressed in local time?

*All times in the article are UTC. We have clarified that where it was ambiguous.*

The authors find very weak correlation between TRT severity rank and lighting activity: it is not clear why a correlation was expected. Moreover, is it the sampling statistically meaningful (only eight severe cells)?

*As stated in the previous answer to comments we wanted to assess whether there was correlation between the two parameters to see whether TRT rank can be used in an operational context. We already suspected that this was not the case and the results shown simply prove that. We think that even though the sampling size is relatively small we already proved the point that TRT rank cannot be used operationally to predict lightning activity.*

At page 9 line 20 "Cells without lightning activity during their life cycle were mostly classified as weak but the rank of the convective cell is a poor indicator of its lightning activity, particularly considering CG flashes." The authors should try to explain the reason for this result.

*We have reformulated the point and provide an attempt at explaining it:*

Cells without lightning activity during their life cycle were indeed classified as weak. However, the rank of the convective cell is a poor indicator of its lightning activity, particularly considering CG flashes. In half of the cells studied, the maximum of lightning activity was reached after the maximum rank was reached and in a quarter it was reached before. Generally speaking, the maximum lightning activity was reached at the time period when cells were classified as weak to moderate. Our hypothesis is that this is linked to the VIL term in the ranking equation, effectively is an integral of reflectivity over height. As such, much more weight is given to the mixed-phase and liquid regions of the precipitating system which, due to the large dielectric constant of their hydrometeors, have a much larger reflectivity. However, it has repeatedly been shown in literature that increases in lightning rate tend to happen before and after the most severe (on the ground) phase of the convective precipitation.

The authors conclude that different ice distribution within clouds is responsible for different lighting efficiency: however, this conclusion is supported by the analysis of only two cells. It is a clue, but for a reliable assessment more cases need to be studied.

*The conclusion is sustained both by the case studies and by the (limited) statistical analysis of the TRT cells. In this particular study we took the chance of having the LMA network to study in more detail intra-cloud lightning activity that is not observable by the operational EUCLID network. We agree that more case studies are needed but we think that nevertheless our observations are worth publishing.*

**Specific comments**

Page 9 line 32 It is not evident why LMA network is usefulness in complex terrain: could the same result be obtained by satellite observation?

*The LMA network provides details of the 3D structure of the intra-cloud lightning. Satellite observations cannot provide that.*

Page 10 lines 1-4 Mosier et al 2010 and Seroka et al 2012 should be taken in account as pioneering works on this topic.

*We added the suggested references to the introduction:*

It should be mentioned that there have already been some attempts at lightning nowcasting based on single-polarization radar products (e.g., Mosier et al., 2011; Seroka et al., 2012).

Figure 1 Quite unclear: please change basemap, add distance reference, North arrow and more contrasting colors.

*We have increased the base map resolution, increased the size of the text and add a north arrow and distance reference as requested. We hope that this satisfies the requests from the reviewer.*

Figure 2 please y-axis range equal to x-axis. 500 LMA observations sound very strange.

*Done as requested. We see nothing strange in the fact that 500 flashes were detected in a TRT cell by the LMA.*

Figure 9 RPC base height equal zero m ASL is quite confusing.

*We changed 0 to NA and mention its meaning in the figure caption.*

Figure 10 please insert a base map with orography in track plots. Please, make y-axis with same range.

*Done as requested*

Figure 11 It is not clear the need to multiply by ten the cell rank. Please, make y-axis with same range

*Done*

Figure 12 The maximum values seem to be spikes (anomaly and isolated very high values)

*The frequency of occurrence is computed over the total number of sources/flashes at each time step, hence if there is just one flash detected over the entire TRT cell volume it will get the value 100%. We have clarified that in the caption:*

Note that the percentage is computed over the total number of flashes/sources at each timestep, i.e. if only one flash was detected over the time step, the corresponding height where the flash was detected will have a value of 100%.

Figure 13 which is the RPC base or top altitude estimations uncertainties? 50 meters?

*The uncertainty is equal to the height resolution, i.e. 250 m*